# An Intricate Network Involving the Argonaute ALG-1 Modulates Organismal Resistance to Oxidative Stress

Carlos A. Vergani-Junior [1,2,9], Raíssa De P. Moro[1,2,9], Silas Pinto [1,2], Evandro A. De-Souza [1], Henrique Camara [1,2,3], Deisi L. Braga[1,2], Guilherme Tonon-da-Silva [1,2], Thiago L. Knittel[1,2], Gabriel P. Ruiz[1,2], Raissa G. Ludwig[1,2], Katlin B. Massirer[4,5], William B. Mair [6] & Marcelo A. Mori [1,2,7,8] ✉

Cellular response to redox imbalance is crucial for organismal health. microRNAs are implicated in stress responses. ALG-1, the *C. elegans* ortholog of human AGO2, plays an essential role in microRNA processing and function. Here we investigated the mechanisms governing ALG-1 expression in *C. elegans* and the players controlling lifespan and stress resistance downstream of ALG-1. We show that upregulation of ALG-1 is a shared feature in conditions linked to increased longevity (e.g., germline-deficient *glp-1* mutants). ALG-1 knockdown reduces lifespan and oxidative stress resistance, while overexpression enhances survival against pro-oxidant agents but not heat or reductive stress. R02D3.7 represses *alg-1* expression, impacting oxidative stress resistance at least in part via ALG-1. microRNAs upregulated in *glp-1* mutants (miR-87-3p, miR-230-3p, and miR-235-3p) can target genes in the protein disulfide isomerase pathway and protect against oxidative stress. This study unveils a tightly regulated network involving transcription factors and microRNAs which controls organisms' ability to withstand oxidative stress.

Aging is associated with the decreased ability of cells to respond to endogenous and exogenous stressors[1,2]. This impaired response to stressful perturbations results in surplus of toxic products, such as reactive oxygen species (ROS), thus impairing health[2]. *Caenorhabditis elegans* is a widely used model organism in aging research and has been fundamental for the identification and characterization of mechanisms involved in life and health span regulation[1,3,4].

One of such mechanisms involve microRNAs (miRNAs) – small non-coding RNAs known to operate by targeting mRNAs through base complementarity and fine-tuning gene expression by inhibiting translation and/or promoting mRNA decay[5]. Since the identification of *lin-4* as a miRNA involved in longevity in *C. elegans*[6], other miRNAs have been found to modulate stress responses and the aging process across the evolutionary spectrum[5,7]. For instance, genetic loss of *mir-71* shortens the lifespan of *C. elegans*, whereas overexpression of this miRNA promotes longevity through pathways involving the transcription factor DAF-16, the worm ortholog of FoxO[8,9]. In addition, *mir-71* mutants fail to properly regulate proteostasis[10].

[1]Department of Biochemistry and Tissue Biology, Institute of Biology, Universidade Estadual de Campinas, Campinas, São Paulo, Brazil. [2]Program in Genetics and Molecular Biology, Institute of Biology, Universidade Estadual de Campinas, Campinas, São Paulo, Brazil. [3]Section on Integrative Physiology & Metabolism, Joslin Diabetes Center, Boston, MA, USA. [4]Center for Molecular Biology and Genetic Engineering (CBMEG), Universidade Estadual de Campinas, Campinas, São Paulo, Brazil. [5]Center of Medicinal Chemistry (CQMED), Universidade Estadual de Campinas, Campinas, São Paulo, Brazil. [6]Department of Molecular Metabolism, Harvard T. H. Chan School of Public Health, Harvard University, Boston, MA, USA. [7]Obesity and Comorbidities Research Center (OCRC), Universidade Estadual de Campinas, Campinas, São Paulo, Brazil. [8]Experimental Medicine Research Cluster (EMRC), Universidade Estadual de Campinas, Campinas, São Paulo, Brazil. [9]These authors contributed equally: Carlos A. Vergani-Junior, Raíssa De P. Moro. ✉e-mail: morima@unicamp.br

Pathways that lead to longevity have been shown to involve at least in part mechanisms that enable organisms to deal more efficiently with stress. For instance, germline deficiency caused by mutation of the *C. elegans* Notch ortholog *glp-1* results in extended lifespan and improved oxidative stress response[11–13]. Interestingly, lifespan extension promoted by the *glp-1* mutation depends on both DAF-16/FoxO and miR-71[9]. The mechanism through which germline deficiency protects from oxidative stress at the organismal level is less understood.

The *C. elegans* argonaute ALG-1 (Ago-like gene 1), like its human ortholog AGO2, is an endoribonuclease that plays a fundamental role in miRNA processing and function[14]. ALG-1 assembles and stabilizes guide strands from miRNA duplexes, pairing them with target mRNAs, thereby mediating target degradation or repression[14]. Inhibition of ALG-1 disrupts miRNA function and leads to a global reduction of mature miRNAs in the worm and accumulation of pre-miRNAs[15,16]. In *C. elegans*, the expression of *alg-1* decreases with aging, and *alg-1* loss-of-function results in developmental defects, as well as decreased oocyte number, shorter lifespan, and increased stress sensitivity[14,16–18]. ALG-1 is also required for proper heat stress recovery response[19]. Although there is mounting evidence showing how miRNAs and argonautes participate in multiple biological processes, including aging, whether these enzymes can be physiologically regulated to modify the ability of organisms to respond to environmental perturbations and adapt to stress remains to be elucidated.

Here we show in *C. elegans* that ALG-1 upregulation is a common feature among a number of conditions leading to extended lifespan. Knockdown of *alg-1* in adult worms reduces lifespan, impairs resistance to pro-oxidant agents, and suppresses the increased longevity and oxidative stress resistance of germline-deficient *glp-1* mutants. In contrast, worms overexpressing *alg-1* are protected from oxidative stress, but not from heat or reductive stress. We identified *R02D3.7* as a repressor of *alg-1* and showed it to be downregulated in *glp-1* mutants and involved in oxidative stress resistance. We also explored the downstream mechanisms through which ALG-1 modulates stress resistance, and found miRNAs upregulated in *glp-1* mutants (i.e., miR-87-3p, miR-230-3p, and miR-235-3p) involved in oxidative stress response. Genes associated with the protein disulfide isomerase (PDI) pathway are enriched among the putative targets of these miRNAs, and we showed that miR-230-3p can directly target *pdi-2* - the ortholog of the human gene encoding protein disulfide isomerase P4HB. Moreover, partial reduction of *pdi-2* results in increased survival when worms are exposed to paraquat – a pro-oxidant agent. Taking these results together, we conclude that upregulation of the argonaute ALG-1 represents a mechanism that helps worms resist oxidative stress by increasing the activity of miRNAs that can target the PDI pathway.

## Results

### Loss of *alg-1* in adult *C. elegans* disrupts redox balance
Deficiency in *alg-1* has a broad range of effects in *C. elegans*, some of which are developmental, i.e., occurring prior to adulthood[15,20]. The levels of ALG-1 reduce with aging[17,21] and *alg-1* deficiency shortens lifespan[17,22]. When *alg-1* RNAi is applied during adulthood to dissociate from its developmental effects, this intervention also reduces lifespan (Supplementary Fig. 1a). While the phenotype of reduced lifespan has been consistently confirmed by other studies[17,22], the mechanisms through which ALG-1 regulates worm vitality and stress resistance remain unclear.

Consistent with an impaired oxidative stress response in *alg-1* deficient worms[18], we found increased mortality in *alg-1* RNAi vs. control RNAi (empty vector) when subjecting worms to pro-oxidant agent paraquat (Supplementary Fig. 1b). *alg-1* RNAi also increased superoxide dismutase 3 (*sod-3*) (Supplementary Fig. 1c) and glutathione S-transferase 4 (*gst-4*) (Supplementary Fig. 1d) promoter activities – known to be induced by oxidative stress. The level of induction of the *sod-3* and *gst-4* transcriptional reporters by *alg-1* RNAi

was equivalent to that induced by paraquat. Moreover, paraquat did not further increase the levels of *sod-3p::GFP* in worms treated with *alg-1* RNAi. These results demonstrate that ALG-1 is necessary in adult worms for proper control of redox homeostasis and oxidative stress resistance.

### ALG-1 is upregulated in multiple long-lived mutants
Given the role of ALG-1 in lifespan and redox homeostasis, we questioned whether *alg-1* expression was increased in long-lived mutants. By analyzing RNA sequencing (RNAseq) data from long-lived worms (GSE111338 from Gene Expression Omnibus and Dutta, et al.[23]), we observed that *alg-1* mRNA was upregulated in the germline-deficient mutant *glp-1(e2141)*, the electron transport chain-deficient mutant *clk-1(qm30)*, the dietary-restricted mutant *eat-2(ad1116)*, and the mTORC1-deficient mutant *raga-1(ok386)*, but not in the insulin/IGF1 signaling-deficient mutant *age-1(hx546)* (Fig. 1a and Supplementary Data 1). On the other hand, *alg-2* - a paralog of *alg-1* also involved in miRNA function[16] - was not affected in these long-lived mutants, except for the *eat-2* mutant, where *alg-2* was increased, and the *glp-1(e2141)* mutant, where *alg-2* was downregulated (Supplementary Fig. 2a).

For that reason, and because the role of ALG-2 in longevity and stress resistance remains contentious[17,18], in this study we decided to focus on ALG-1. Using a CRISPR-engineered ALG-1 reporter strain generated by the Pasquinelli group where GFP was fused to the N-terminus of endogenous ALG-1[17] (PQ530), we confirmed that the upregulation of ALG-1 occurred at the protein level in all mutants where *alg-1* mRNA was upregulated (Fig. 1b, c). The *glp-1(e2144)* mutant exhibited the highest ALG-1 expression among the long-lived mutants. Thus, we used *glp-1(e2144)* mutants for the remainder of the study to better comprehend ALG-1 regulation and its association with stress resistance in a condition of extended longevity.

To understand how ALG-1 expression can be transcriptionally controlled and dynamically regulated, we searched for transcription factors that bind to the *alg-1* promoter and control its expression. Using the modENCODE project - a validated ChIP-Seq database of *C. elegans*[24] - we found 24 transcription factors capable of binding to a 2-kb element upstream of the *alg-1* starting codon (Fig. 1d and Supplementary Data 2). First, we tested the ability of these transcription factors to control ALG-1 levels using an RNAi screen employing a multicopy, integrated *alg-1p::gfp::alg-1* transgenic reporter strain generated by Shih-Peng Chan & Frank Slack (CT20)[25].

Of the 24 candidates, knockdown of 5 of them (*gei-2/mep-1, nhr-28, nhr-77, R02D3.7,* and *skn-1*) increased the levels of ALG-1 (Fig. 1e). To further validate these candidates as regulators of *alg-1*, we used the CRISPR-engineered ALG-1 reporter strain PQ530. In PQ530 worms, ALG-1 levels increased following the knockdown of *gei-2, nhr-77,* or *R02D3.7*, with no changes observed under *nhr-28* and *skn-1* RNAi (Fig. 1f). These results confirm GEI-2, NHR-77, and R02D3.7 as negative regulators of ALG-1.

We then investigated the expression levels of *gei-2, nhr-77,* and *R02D3.7* in the RNAseq datasets of long-lived mutants. We found that *R02D3.7* was downregulated in all long-lived mutants where *alg-1* was upregulated (Fig. 1g), agreeing with its role as a negative regulator of *alg-1*. The pattern of *gei-2* was less consistent, whereas it was also downregulated in *glp-1(2144)* mutants. In contrast, *nhr-77* was not altered in the long-lived mutants (Fig. 1g and Supplementary Data 1). Knockdown of *R02D3.7*, but not *gei-2*, further increased ALG-1 expression in *glp-1(e2144)* mutants (Supplementary Fig. 2b), although to a lesser magnitude than in wild type (WT) worms (Fig. 1e). Together, these results identify three genes that exert negative regulation over *alg-1*; two of them, namely *R02D3.7* and *gei-2*, are associated with the upregulation of ALG-1 in long-lived mutants.

While the binding of these transcription factors to the *alg-1* promoter suggests their *cis*-regulatory role in controlling *alg-1* transcription, the verification of their functionality in a screen utilizing reporter

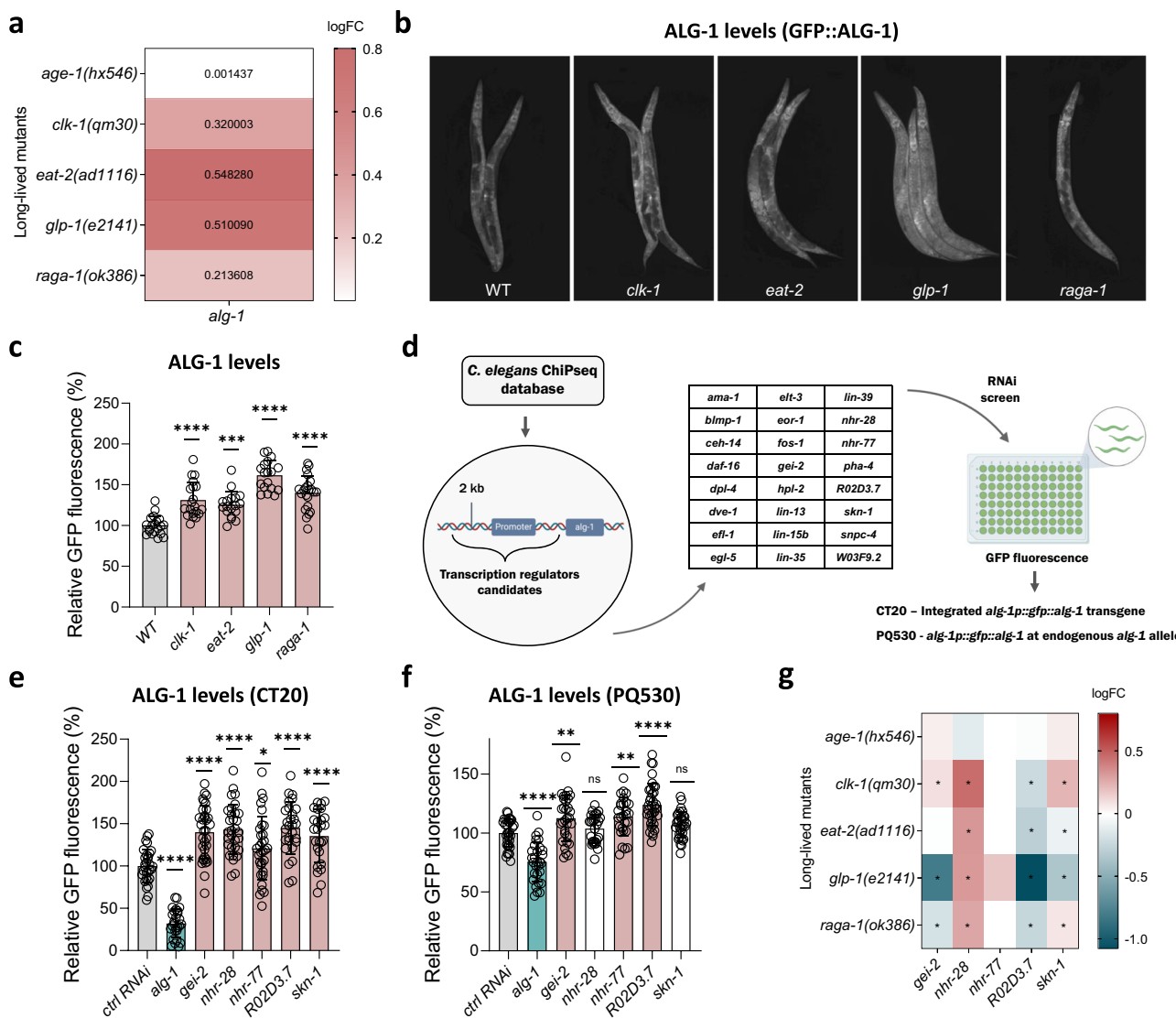

**Fig. 1 | ALG-1 levels are increased in long-lived mutants. a** *alg-1* gene expression in long-lived mutants as determined by RNAseq. $n = 4$ per group. Values represent $\log_2$(fold change of long-lived mutant/control), logFC. Adjusted *P* values and detailed information are described in Supplementary Data 1. **b** Representative images and **c** relative GFP fluorescence of the *gfp::alg-1* reporter (PQ530) in different long-lived mutant backgrounds at day 1 of adulthood. Combined data of two independent replicates. $n = 20, 20, 18, 18, 22$ worms per condition. **d** Schematic of the RNAi screen used to find and validate *alg-1* regulators. Two *gfp::alg-1* reporter strains were used, CT20[25], a multicopy integrated *alg-1p::gfp::alg-1* transgene (*zals5*) and, PQ530[17], a endogenous *alg-1* allele (*ap423*) tagged with GFP. Created with BioRender.com. Relative GFP fluorescence of CT20 ($n = 30, 27, 33, 31, 31, 28, 26$

worms per condition) (**e**) and PQ530 ($n = 32, 29, 31, 28, 28, 35, 36$ worms per condition) (**f**) worms after RNAi exposure. Worms were grown on plates with control (ctrl) RNAi (empty vector), transferred to respective RNAi plates at day 0 of adulthood, and had their fluorescence measured on day 3. Combined data of three independent experiments. **g** *gei-2/mep-1*, *nhr-28*, *nhr-77*, *RO2D3.7* and *skn-1* gene expression in long-lived mutants as determined by RNAseq. $n = 4$ per group. * Adjusted *P* values < 0.05, as described on Supplementary Data 1. **c, e, f** Bars represent mean ± SEM. Comparisons between controls groups *versus* experimental groups were made using one-way ANOVA with Dunnett's post hoc test. ns – non-significant ($P > 0.05$), * $P < 0.05$, **$P < 0.01$, ***$P < 0.001$, ****$P < 0.0001$. Source data and exact *P* values (whenever available) are provided in the Source Data file.

strains where the *alg-1* promoter governs the expression of GFP::ALG-1 fusion proteins potentially influenced by the endogenous 5' and 3' UTRs does not preclude the possibility of their acting in *trans*. For instance, they might exert an indirect influence through the induction of miRNAs that suppress ALG-1 expression. Indeed, the long-lived phenotype of *glp-1* mutants depends on miR-71[9], which is itself a negative regulator of ALG-1[21]. Hence, the reduction of a positive regulator of miR-71 could explain the upregulation of ALG-1.

In agreement with this possibility, a null mutation in the *mir-71* gene upregulated ALG-1 in the ALG-1 reporter strain CT20 (Supplementary Fig. 2c). We then tested if *gei-2* and *RO2D3.7*, which are downregulated in *glp-1* mutants and suppress ALG-1 levels, control the activity of the *mir-71* promoter (Supplementary Fig. 2d). Only the

knockdown of *gei-2* influenced *mir-71* promoter activity, resulting in downregulation. Consistent with a function of GEI-2 suppressing ALG-1 via induction of miR-71, *gei-2* RNAi did not induce GFP::ALG-1 levels in *mir-71* mutant worms (Supplementary Fig. 2e). These findings unveil a complex regulatory network governing ALG-1 levels, wherein transcription factors can modulate ALG-1 expression both in *cis* through direct promoter binding and in *trans* by inducing miRNAs that negatively regulate its mRNA.

## Worms overexpressing ALG-1 are oxidative stress resistant

To investigate whether upregulation of ALG-1 is sufficient to prolong worm lifespan and protect adult worms from stress, we studied worms overexpressing *alg-1* (ALG-1/OE) (Supplementary Fig. 3a–d). One of

these models carries a multicopy transgene inserted in the genome from which a GFP::ALG-1 fusion functional protein is overexpressed (CT20)[25] – the same model used as one of the ALG-1 reporter strains in the previous experiments (Fig. 1d). In this case, to control for the transgene expression, we compared these worms to worms carrying a transgene overexpressing a luciferase::GFP fusion protein (PE255), a strain generated by Cristina Lagido[26], which we have denoted as the ctrl strain.

We also made use of another ALG-1/OE model, in which the 3'UTR of *alg-1* was substituted by the *Y45F10D.4* 3'UTR using CRISPR/Cas9 editing (PQ535), thus avoiding the negative regulation exerted by miRNAs, as designed, and previously reported by the Pasquinelli lab[17]. In this case, we used the WT as the control. Both models exhibited at least 4-fold increases in *alg-1* mRNA levels in comparison to their respective controls (Supplementary Fig. 3b and d).

While none of the ALG-1/OE strains lived longer or had a consistent delay in the progression of age-related features such as the decline in mobility and pharyngeal pumping capacity (Supplementary Fig. 4a–f), the PQ535 strain exhibited delayed duration of development (Supplementary Fig. 4g) and reduced brood size (Supplementary Fig. 4h) when compared to WT worms, in agreement with the crucial role of miRNAs in development and reproduction[27–29].

However, bacterial-derived nitric oxide (NO), a free radical known to inhibit ALG-1 post-translationally[30] and to play a role in worm redox reactions and signaling pathways[31–34], influences ALG-1/OE effects on *C. elegans* lifespan, as worms overexpressing ALG-1 lived longer than controls when fed with *B. subtilis* (ΔNOS) that do not produce NO (Supplementary Fig. 4i). In contrast, and like *E. coli*, in the presence of wild type *B. subtilis* ALG-1/OE worms lived as long as the controls. These phenotypes can be attributed to a decrease in lifespan observed when worms are exposed to bacteria lacking NO, whereas ALG-1 overexpression confers protection against this lifespan reduction. Noteworthy, NO reacts with superoxide to produce peroxynitrite, and in NO deficient conditions, increased superoxide levels and an over-reliance on superoxide dismutases to maintain redox balance can be anticipated[35].

In agreement with a protective role of ALG-1 overexpression under oxidative stress conditions, ALG-1/OE worms displayed increased resistance against sodium arsenite and paraquat stresses, regardless of the model to overexpress ALG-1 (Fig. 2a–d). Both chemicals are known to enhance superoxide levels, albeit through different mechanisms[36–38]. To control for a potential unspecific effect of GFP overexpression, we silenced GFP in the ALG-1/OE (CT20) model and this completely reversed the paraquat resistance phenotype of these worms, consistent with the silencing of the transgene-driven GFP::ALG-1 fusion protein. In contrast, silencing GFP did not affect paraquat sensitivity in the control PE255 strain, which overexpresses the luciferase::GFP fusion protein (Supplementary Fig. 5a).

Germline-deficient *glp-1* mutant worms were also more resistant to sodium arsenite and paraquat (Fig. 2e, f), although in the arsenite assay the differences did not reach statistical significance (P = 0.0576 at 15 mM). Furthermore, increased paraquat resistance and longevity of *glp-1* mutants required ALG-1 (Fig. 2f and Supplementary Fig. 5b). These findings support the notion that elevated levels of ALG-1 in *glp-1* mutants contribute to enhanced resistance against oxidative stress, and further establish the indispensability of ALG-1 for the prolonged survival of germline-deficient worms in both normal and stressful environmental conditions.

The transcription factors DAF-16 and SKN-1 play a role in multiple stress pathways and are required for stress resistance and longevity induced by germline deficiency[13,39–41]. This prompted us to investigate whether ALG-1 acted in concert with DAF-16 and SKN-1 to promote oxidative stress resistance. While RNAi targeting *daf-16* or *skn-1* reduced worm survival under paraquat, ALG-1 overexpression increased worm survival regardless whether *daf-16* and *skn-1* were

knocked down or not (Supplementary Fig. 5c, d), indicating that ALG-1 induction promotes oxidative stress resistance despite the silencing of these transcription factors.

To further explore how ALG-1 upregulation protects from oxidative stress, we evaluated the promoter activity of genes encoding two enzymes involved in antioxidant and detoxification responses – SOD-3 and GST-4. Improved oxidative stress resistance in ALG-1/OE worms accompanied higher *sod-3* promoter activity levels at baseline (Fig. 2g), but not *gst-4* (Fig. 2h). Paraquat induced *sod-3p::GFP* and *gst-4p::GFP* levels in both ALG-1/OE and WT worms, whereas the activity of the *sod-3* reporter was higher in ALG-1/OE under this oxidative stress condition (Fig. 2g, h).

The administration of the antioxidant N-acetylcysteine (NAC) blunted the induction of the *sod-3* reporter by *alg-1* RNAi, but had no impact on WT or ALG-1/OE worms (Fig. 2i). Hence, ALG-1/OE worms remained with higher levels of the *sod-3* reporter in comparison to WT worms regardless of the presence of NAC. This suggests that, in contrast with *alg-1* knockdown where both *sod-3* and *gst-4* promoters are induced due to a chronic state of oxidative stress, overexpression of ALG-1 selectively induces *sod-3* transcription, thus helping worms to resist superoxide exposure in a preventive manner.

Importantly, the increased stress resistance promoted by ALG-1 overexpression was specific to oxidative stress, given that ALG-1/OE worms were less resistant to dithiothreitol (DTT) - a reducing agent known to induce endoplasmic reticulum (ER) stress[42,43] (Supplementary Fig. 5e) – and as resistant as the controls to heat stress (Supplementary Fig. 5f). Overall, these results show that ALG-1 overexpression is sufficient to enhance worm survival under conditions of oxidative stress.

## Downregulation of *R02D3.7* protects from oxidative stress

Next, we investigated if the two negative regulators of ALG-1 that are downregulated in *glp-1* mutants (i.e., *R02D3.7* and *gei-2*) control oxidative stress response in a manner that involved ALG-1. While *R02D3.7* RNAi protected worms from sodium arsenite and paraquat-induced mortality, the knockdown of *gei-2* had no significant effect on oxidative stress resistance (Fig. 3a, b). Adding *alg-1* RNAi to the *R02D3.7* RNAi fully suppressed the effect of *R02D3.7* silencing on arsenite stress (Fig. 3c). However, *R02D3.7* RNAi could still increase protection from paraquat stress when *alg-1* was knocked down, whereas the addition of *alg-1* RNAi decreased paraquat resistance when *R02D3.7* was silenced (Fig. 3d).

Furthermore, in contrast to the observed effects on WT worms, the knockdown of *R02D3.7* showed limited impact on ALG-1/OE worms, failing to further increase resistance to paraquat (Fig. 3e). Consistent with the normal lifespan of ALG-1/OE worms, *R02D3.7* knockdown did not affect survival under unstressed conditions (Fig. 3f). Together, these results show that the downregulation of *R02D3.7*, which allows higher levels of ALG-1, protects worms from oxidative stress. This phenotype occurs at least partially via ALG-1 and can be mimicked by ALG-1 overexpression.

## miRNAs induced by ALG-1/OE confer oxidative stress resistance

We then searched for miRNAs that conferred improved oxidative stress resistance to worms overexpressing ALG-1. Since miR-71 is required for germline deficiency-induced longevity[9], we first investigated whether miR-71 deficiency interfered with the stress resistance of ALG-1/OE worms. Loss of *mir-71* blunted the protection against paraquat (Supplementary Fig. 6a) and sodium arsenite (Supplementary Fig. 6b) promoted by ALG-1/OE, although it did not interfere with oxidative stress resistance of control worms (Supplementary Fig. 6c, d). These results indicate that in addition to being a negative regulator of ALG-1[21], miR-71 also acts downstream of ALG-1 overexpression to confer increased oxidative stress resistance.

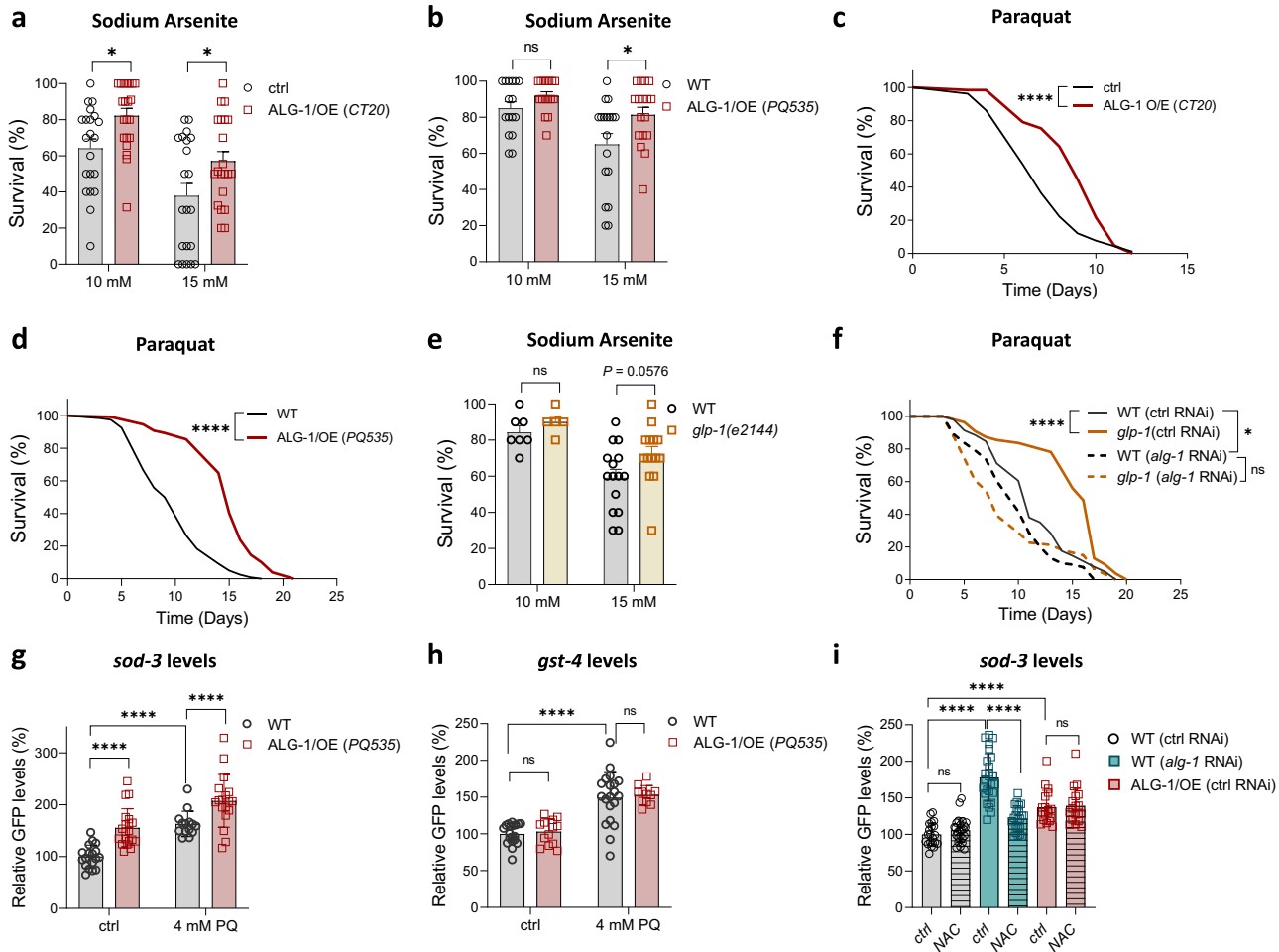

**Fig. 2 | *alg-1* overexpression improves oxidative stress resistance.** Survival of day 3 adults after 8 hours on sodium arsenite of **a** CT20 (*n* = 21 wells with 10 worms each per condition) and **b** PQ535 (*n* = 16, 18, 16, 17 wells with 10 worms each per condition) strains in comparison with their controls. Combined data from three independent experiments. Survival on paraquat (4 mM) of **c** CT20 (*n* = 106 worms for ctrl; 128 worms for CT20) and **d** PQ535 (*n* = 122 worms for WT; 161 worms for PQ535) in comparison with their controls. Representative data from three independent replicates. **e** Survival of day 3 *glp-1(e2144)* and WT adult worms after 8 hours on sodium arsenite. *n* = 7, 5, 15, 15 wells with 10 worms each per condition. Combined data from two independent replicates. **f** Survival on paraquat (8 mM) of *glp-1(e2144)* and WT worms treated with *alg-1* or control (ctrl) RNAi (empty vector) from day 0. *n* = 115 worms for WT on ctrl RNAi; 151 worms for WT on *alg-1* RNAi; 131 worms for *glp-1(e2144)* on ctrl RNAi; 120 worms for *glp-1(e2144)* on *alg-1* RNAi. Representative

data from two independent replicates. Relative GFP fluorescence of day 3 worms expressing **g** *sod-3p::gfp* (*muIs84*) (*n* = 18, 20, 14, 18 worms per condition) or **h** *gst-4p::gfp* (*dvIs19 III*) (*n* = 19, 13, 20, 11 worms per condition) reporters on the WT or ALG-1/OE (PQ535) backgrounds. Combined data from two independent replicates. **i** Relative GFP fluorescence of day 3 worms expressing the *sod-3p::gfp* (*muIs84*) reporter under *alg-1* RNAi or further carrying the ALG-1/OE transgene (PQ535). Worms were treated with 10 mM N-acetylcysteine (NAC) or vehicle (ctrl). *n* = 21, 25, 26, 22, 18, 18 worms per condition. Combined data from two independent replicates. **c, d** and **f** Data were compared using the log-rank test. ns – non-significant (*P* > 0.05), *\*P* < 0.05, *\*\*\*\*P* < 0.0001. **a, b, e, g–i** Bars represent mean ± SEM. Comparisons were made using two-way ANOVA with Sidak's post hoc test. ns – non-significant (*P* > 0.05), *\*P* < 0.05, *\*\*\*\*P* < 0.0001. Source data and exact *P* values (whenever available) are provided in the Source Data file.

Next, we took an unbiased approach and performed small RNAseq to identify potential miRNAs commonly upregulated in *glp-1(e2144)* mutants and ALG-1/OE (CT20) worms. Consistent with increased ALG-1 levels in both models, we saw an overall increase in miRNA abundance and a significant overlap (*P* < 0.02) between the miRNAs upregulated in the two conditions (Fig. 4a–d and Supplementary Data 3). There were 72 miRNAs upregulated in *glp-1(e2144)* mutants and 36 in ALG-1/OE (CT20) (log₂ fold change > 1), among which 11 were in common (Fig. 4c, e). In contrast, only 22 miRNAs were downregulated in *glp-1(e2144)* mutants and 4 in ALG-1/OE (CT20) (log₂ fold change < −1), with no overlap (Fig. 4d). Although miR-71 did not fall within the miRNAs affected by ALG-1 overexpression according to our cutoff, it appeared increased in *glp-1(e2144)*.

Of the 11 upregulated miRNAs identified in *glp-1(e2144)* mutants and ALG-1/OE (CT20) worms, both the 5p and 3p strands of miR-87 and miR-230 were observed, contributing to a total of 9 potential miRNA genes associated with stress resistance (Fig. 4e). Among those, we had access

to null mutants of 8 of them (i.e., *mir-45*, *mir-47*, *mir-87*, *mir-229*, *mir-230*, *mir-235*, *mir-259*, *mir-357*). Initially, we tested if these mutants were more sensitive to oxidative stress. *mir-235* deficiency impaired both arsenite and paraquat stress resistance (Fig. 4f, i), while loss of *mir-230* or *mir-87* impaired only paraquat-induced mortality (Fig. 4g, h).

Loss of *mir-229*, on the other hand, improved paraquat stress resistance (Supplementary Fig. 7j). The null mutants for the remaining miRNAs did not show any differences in the resistance to oxidative stress (Supplementary Fig. 7a–l). Loss-of-function mutations in *mir-87*, *mir-230*, or *mir-235* also led to reduced oxidative stress resistance in ALG-1/OE worms (Supplementary Fig. 8a–f). These findings unveiled miRNAs upregulated in *glp-1* mutants and ALG-1/OE worms that take part in regulation of oxidative stress resistance.

**miRNAs induced by ALG-1/OE target and limit the PDI pathway**
miR-235, miR-230, and miR-87 were the miRNAs that consistently conferred protection against oxidative stress-induced mortality

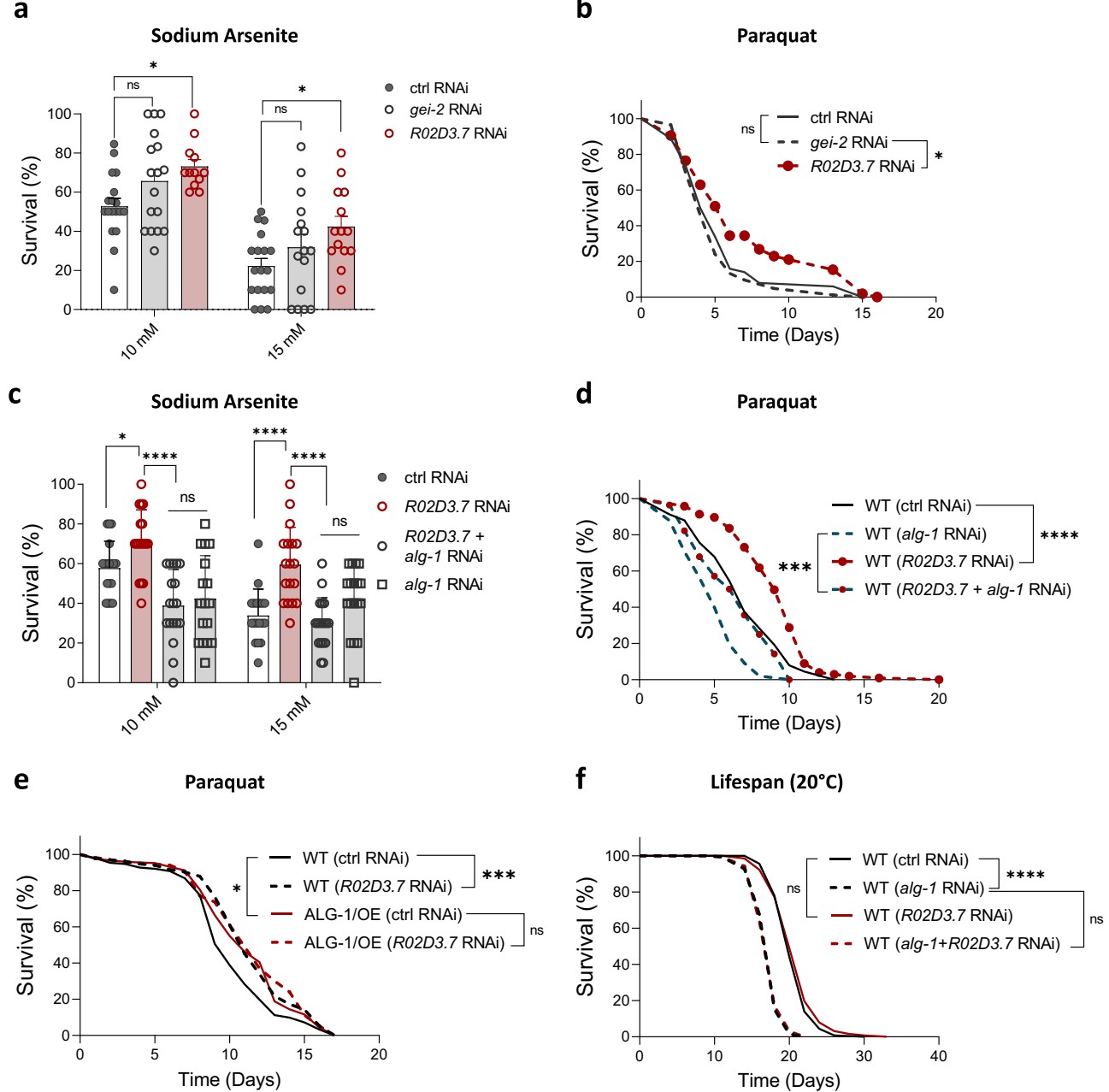

**Fig. 3 | Knockdown of *R02D3.7* increases oxidative stress resistance. a** Survival after 8 hours of sodium arsenite exposure on day 3 adults. WT worms were treated with control (empty vector), *gei-2* or *R02D3.7* RNAi since day 0. *n* = 18, 17, 12, 18, 16, 14 wells with 10 worms each per condition. **b** Survival on paraquat (8 mM) of WT worms treated with control (empty vector), *gei-2* or *R02D3.7* RNAi since day 0 of adulthood. *n* = 98 worms for ctrl RNAi; 113 worms for *gei-2* RNAi; 112 worms for *R02D3.7* RNAi. **c** Survival after 8 hours of sodium arsenite exposure on day 3 adults. WT worms were treated with control (empty vector), *R02D3.7*, *R02D3.7* + *alg-1* or *alg-1* RNAi since day 0. *n* = 18 wells with 10 worms each per condition. **d** Survival on paraquat (8 mM) of WT worms treated with control (empty vector), *R02D3.7*, *R02D3.7* + *alg-1* or *alg-1* RNAi since day 0 of adulthood. *n* = 91 worms for ctrl RNAi; 99 worms for *alg-1* RNAi; 117 worms for *R02D3.7* RNAi; 78 worms for *R02D3.7*+*alg-1* RNAi. **e** Survival on paraquat (8 mM) of WT and ALG-1/OE (PQ535) worms treated

with control (empty vector) or *R02D3.7* RNAi since day 0 of adulthood. *n* = 152 worms for WT ctrl RNAi; 133 worms for WT *R02D3.7* RNAi; 124 worms for ALG-1/OE ctrl RNAi; 118 worms for ALG-1/OE *R02D3.7* RNAi. **f** Lifespan under normal conditions of WT worms treated with control (empty vector), *R02D3.7*, *R02D3.7* + *alg-1* or *alg-1* RNAi since day 0 of adulthood. *n* = 136 worms for ctrl RNAi; 132 worms for *alg-1* RNAi; 127 worms for *R02D3.7* RNAi; 135 worms for *R02D3.7*+*alg-1* RNAi. **a, c** Bars represent mean ± SEM. Combined data of three independent replicates. Comparisons were made using two-way ANOVA with Sidak's post hoc test. ns – non-significant (*P* > 0.05), **P* < 0.05, *****P* < 0.0001. **b, d–f** Representative data from three independent replicates. Data were compared using the log-rank test. ns – non-significant (*P* > 0.05), **P* < 0.05, ****P* < 0.001, *****P* < 0.0001. Source data and exact *P* values (whenever available) are provided as a Source Data file.

according to our data. Hence, we decided to look for targets of these miRNAs that participate in oxidative stress resistance. We intersected the information of putative targets of these miRNAs (we used target predictions of the 3p strands as it was the information available on the TargetScanWorm database[44]) with possible miRNA targets, i.e., genes

found to be upregulated in publicly available microarray data obtained from the *alg-1* loss of function mutant *alg-1(gk214)*[45] (Fig. 5a).

Out of the 581 genes upregulated in *alg-1(gk214)* mutants, 41 (7%, *P* = 7.33e-05) overlapped with 344 putative targets of miR-87-3p, whereas for miR-230-3p and miR-235-3p, there was an overlap of 137

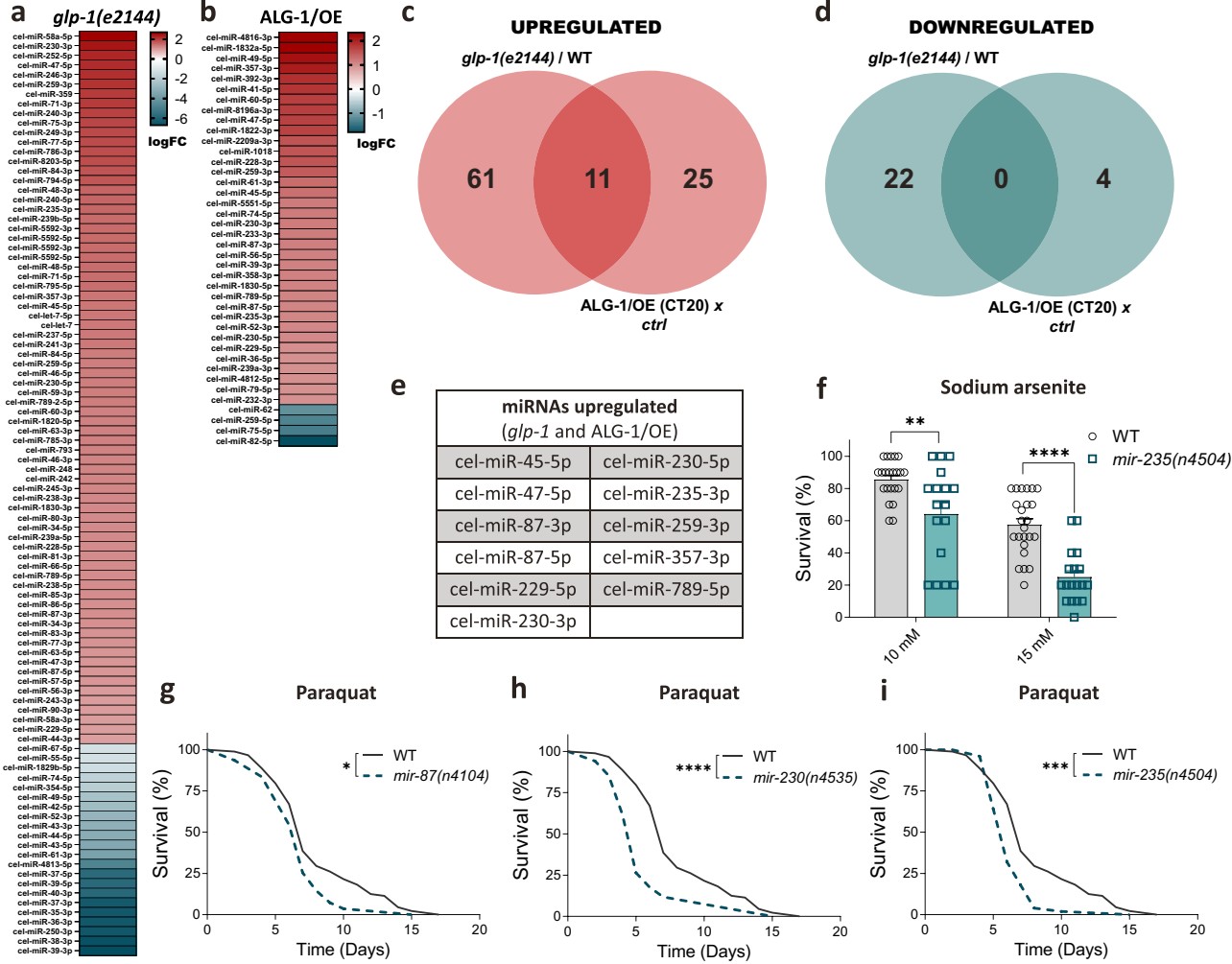

**Fig. 4 | Increased miRNAs upon ALG-1 overexpression control resistance to oxidative stress.** Heatmaps of differentially expressed miRNAs ($\log_2$ fold change > 1 or < −1) on **a** a *glp-1(e2144)* mutants in comparison with WT worms, and **b** ALG-1/OE (CT20) strain in comparison with its control (ctrl) strain. Detailed information in Supplementary Data 3. Venn diagrams representing the miRNAs (**c**) upregulated or (**d**) downregulated on *glp-1(e2144)*, ALG-1/OE (CT20), and in both. **e** Eleven miRNAs upregulated in both *glp-1(e2144)* mutants and ALG-1/OE worms. **f** Survival of *mir-235(n4504)* and WT worms after 8 hours of exposure to sodium arsenite on day 3 of adulthood. $n = 22, 17, 24, 18$ wells with 10 worms each per condition. Bars represent mean ± SEM. Combined data of three independent replicates. Comparisons were made using two-way ANOVA with Sidak's post hoc test. \*\**P* < 0.01, \*\*\*\**P* < 0.0001. **g**–**i** Survival on paraquat (4 mM) of *mir-87(n4104)*, *mir-230(n4535)* or *mir-235(n4504)* mutants *vs.* WT. Experiments were performed together and share the same controls, which were split in different panels to allow better visualization. $n = 119$ worms for WT; 133 worms for *mir-87*; 110 worms for *mir-230*; 123 worms for *mir-235*. Data were compared using the log-rank test. \**P* < 0.05, \*\*\**P* < 0.001, \*\*\*\**P* < 0.0001. **f**–**i** Data represent one experiment of three independent replicates. Source data are provided as a Source Data file.

(23.5%, $P = 6.32e{-}60$) and 36 (6%, $P = 8.07e{-}07$) genes of a total of 1,022 and 223 putative targets, respectively (Supplementary Fig. 9a and Supplementary Data 4). According to the GO Molecular Function database, available at WormEnrichR[46], these overlapped genes are predicted to have their molecular function associated with the PDI pathway (Supplementary Fig. 9b).

Based on this data, we hypothesized that miRNA-mediated targeting of the PDI pathway could help worms to better resist oxidative stress. Considering that the mRNA of one of the *C. elegans* PDI isoforms, PDI-2, appeared as a putative direct target of miR-230-3p (Supplementary Data 4), we attempted to confirm this mechanism using a reductionist experimental setting. We transfected HEK293T cells with luciferase reporter vectors carrying the *pdi-2* 3′UTR (native or mutated) and added the *C. elegans* miR-230-3p mimic (Fig. 5b). The miR-230-3p mimic inhibited luciferase activity in cells transfected with the vector containing the native *pdi-2* 3′UTR, but not in cells transfected with the mutated vector (Fig. 5c), confirming that *pdi-2* can indeed be a direct target of miR-230-3p.

Furthermore, the partial inhibition of *pdi-2* in the *pdi-2(tm689)* heterozygous loss-of-function mutant protects worms from paraquat or sodium arsenite exposure (Fig. 5d, e). Concordantly, using RNAi to downregulate *pdi-2* increased paraquat resistance (Fig. 5f). On the other hand, this approach sensitized worms to sodium arsenite stress (Fig. 5g) and reduced lifespan under normal conditions (Fig. 5h). Taken together, our results demonstrate that upregulation of ALG-1, which occurs in multiple long-lived mutants, enhances oxidative stress resistance by upregulating miRNAs that can target and fine-tune the PDI pathway (Fig. 6).

## Discussion

Aging progressively decreases the ability of cells to respond to stressors. Hence, being able to maintain stress response mechanisms can prolong the general health span of the organism[1,2]. Several individual miRNAs participate in pathways directly involved in aging and stress responses[5,7]. In the current study, we described a central pathway through which *C. elegans* modulates the miRNA processing machinery to resist oxidative stress.

**Fig. 5 | Fine-tuning of the PDI pathway by miRNAs improves oxidative stress resistance. a** Schematic of the strategy used to find putative targets of the miRNAs upregulated in response to *alg-1* overexpression and involved in oxidative stress resistance. **b** Approach used to validate *pdi-2* as a target of miR-230-3p. Cel-miR-230-3p or cel-miR-85-3p (negative control) mimics were administrated in HEK293T cells transfected with luciferase reporter vectors containing the *pdi-2* 3'UTR (native or mutated – Firefly luciferase) or not (Renilla luciferase). Created with BioRender.com **c** Luminescence ratio of Firefly luciferase (potentially regulated by the presence of *pdi-2* 3'UTR) and Renilla luciferase (transfection control) of HEK293T cells. *n* = 12 independent cell pools per condition. **d** Survival of *pdi-2(tm689)* heterozygous (het) mutants and WT worms on paraquat (4 mM). *n* = 122 for WT and 85 for *pdi-2(tm689)*. **e** Survival of *pdi-2(tm689)* heterozygous (het) mutants and WT worms after 8 hours of exposure to sodium arsenite on day 3 of

adulthood. *n* = 22, 22, 28, 28 wells with 10 worms each per condition. **f** Survival on paraquat (8 mM) of WT worms exposed to control (ctrl, empty vector) or *alg-1* RNAi since eggs. *n* = 91 worms for ctrl RNAi and 131 worms for *pdi-2* RNAi. **g** Survival after 8 hours of exposure to sodium arsenite on day 3 of adulthood of WT worms exposed to ctrl (empty vector) or *pdi-2* RNAi since eggs. *n* = 16, 16, 20, 20 wells with 10 worms each per condition. **h** Lifespan of WT worms exposed to ctrl (empty vector) or *alg-1* RNAi since eggs. *n* = 142 worms for ctrl RNAi and 146 worms for *pdi-2* RNAi. **c, e, g** Bars represent mean ± SEM. Combined data from two independent experiments. Comparisons were made using two-way ANOVA with Sidak's post hoc test. ns – non-significant (*P* > 0.05), ****P* < 0.0001. **d, f** and **h** Data were compared using the log-rank test. *****P* < 0.0001. Representative data from three independent experiments. Source data and exact *P* values (whenever available) are provided as a Source Data file.

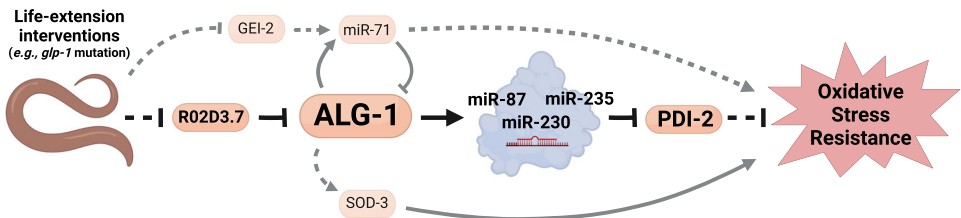

**Fig. 6 | An intricate mechanism involving the miRNA biogenesis machinery regulates oxidative stress resistance in *C. elegans*.** Arrows represent activation. Blunt-end represents inhibition. Continuous lines represent direct regulation. Dashed lines represent indirect regulation. Created with BioRender.com.

We used *C. elegans* as a model to characterize this pathway since the miRNA processing machinery is highly conserved from nematodes to humans, and mechanisms involved in organismal stress responses and aging are well established in *C. elegans* and can be more easily assessed in this species. In addition, age-associated changes in components of the miRNA processing machinery such as Dicer and argonautes have been demonstrated in *C. elegans* and confirmed in vertebrates[17,47]. While upregulation of Dicer in *C. elegans* renders

worms stress resistant[47], studies in multiple organisms suggest that the levels of argonautes are usually limiting when it comes to availability of most miRNAs[16,48]. Indeed, argonaute loading has been recently confirmed to represent a key bottleneck for miRNA stabilization[49]. Moreover, Dicer processes different small RNA species, while argonautes such as human AGO2 and *C. elegans* ALG-1 are specific to miRNA[14]. Hence, targeting argonautes may represent a more specific and efficacious way to alter miRNA activity.

While ALG-2, along with ALG-1, contributes to miRNA function and biogenesis in *C. elegans*, its role in lifespan regulation and stress resistance is less clear. Reductions in lifespan and in stress resistance in *alg-1* loss-of-function mutants have been confirmed by different groups, whereas *alg-2* loss-of-function mutants have been reported as long-lived or short-lived in different studies[17,18]. Moreover, *alg-1* loss-of-function mutants are more sensitive to oxidative stress than *alg-2* loss-of-function mutants[18]. In addition, our findings indicate that while *alg-1* levels are induced in several long-lived strains, the levels of *alg-2* are not consistently related to longevity in these models.

In line with previous findings[17,18,22], we show that knocking down *alg-1* in adult worms reduces lifespan and impairs resistance to pro-oxidant agents. These phenotypes are accompanied by increases in *sod-3* and *gst-4* promoter activities in *alg-1* deficient worms. Interestingly, the level of induction of these oxidative stress sensors in worms treated with *alg-1* RNAi is like that obtained after exposure to paraquat, and the level of the *sod-3* promoter activity in worms where *alg-1* is knocked down is reduced in the presence of the antioxidant NAC, agreeing with a chronic state of oxidative stress in these worms.

One hypothesis to explain the overall dysfunction in redox homeostasis in adult worms deficient in *alg-1* is the fact that DAF-16 activity is inhibited when ALG-1 function is lost[17]. DAF-16 orchestrates multiple stress response pathways in *C. elegans*, including oxidative stress response[50,51], and is involved in various longevity pathways such as those triggered by defects in insulin/IGF1 signaling or germline ablation[52–54]. DAF-16 inhibition could render worms more prone to suffer from oxidative stress and thereby have a shorter lifespan. However, *alg-1* deficiency further decreases the lifespan of *daf-16* loss-of-function mutants[17]. In addition, most of the genes differentially expressed in *alg-1(gk214)* mutants are upregulated (consistent with a defect in miRNA biogenesis/function) and are not targets of DAF-16[17], indicating that there are additional players downstream of ALG-1 that may participate in stress regulation and longevity.

While loss-of-function models of *alg-1* can provide valuable insights into the fundamental role of ALG-1 in maintaining overall health, *alg-1* deficiency is expected to disrupt multiple biological processes, and increased stress might reflect these widespread alterations, making it difficult to pinpoint how changes in ALG-1 physiologically control aging and stress response. Therefore, we searched conditions in which ALG-1 expression is induced and thus used gain-of-function models to investigate how ALG-1 upregulation affects longevity and stress resistance. We observed that ALG-1 is upregulated in various long-lived mutants [i.e., *glp-1(e2141)*, *clk-1(qm30)*, *eat-2(ad1116)*, and *raga-1(ok386)*], but not in the insulin/IGF1 signaling deficient mutant *age-1(hx546)*. Intermittent fasting, another life-extension intervention, also leads to *alg-1* induction in *C. elegans*[55].

Guided by the modENCODE data, we performed RNAi screen on *alg-1::gfp* reporter strains to validate *alg-1* regulators. We found that the transcription regulators GEI-2/MEP-1, NHR-77, and R02D3.7 negatively control ALG-1 levels in two distinct reporter strains. However, while *nhr-28* or *skn-1* RNAi was found to derepress *alg-1* in the CT20 ALG-1 reporter strain (generated by transgene insertion), it did not affect ALG-1 levels in the PQ530 strain (generated by CRISPR engineering). Hence, RNAi targeting *nhr-28* or *skn-1* could be influencing ALG-1 levels in the CT20 strain via unspecific mechanisms (for instance, inhibition of transgene silencing[56]).

Among the genes encoding these transcription regulators, only *R02D3.7* is consistently downregulated in all long-lived mutants where *alg-1* is upregulated, suggesting its importance as a key factor in controlling *alg-1* expression in the context of longevity. Corresponding with its role in oxidative stress resistance, the knockdown of *R02D3.7* enhances survival in sodium arsenite and paraquat stress assays. Notably, the impact of R02D3.7 on arsenite survival is entirely reliant on ALG-1, while its contribution to paraquat resistance appears to extend beyond ALG-1. Conversely, like the observed effects upon ALG-1 overexpression, RNAi targeting *R02D3.7* fails to extend lifespan under normal conditions. R02D3.7 is a transcription factor carrying the well-conserved C2H2-type zinc finger motif. A study has recently described R02D3.7 as a negative regulator of *cat-1* and *cat-2* – genes involved in dopamine metabolism in *C. elegans* - rendering it the name RCAT-1 (regulator of cat genes)[57]. Interestingly, R02D3.7 was identified as responsive to the pheromone ascaroside, which normally serves as a signal of starvation or overcrowding in *C. elegans*[57]. Altogether, these results point to R02D3.7/RCAT-1 as a stress sensor controlling *alg-1* transcription and stress response in *C. elegans*.

The *alg-1* transcript can also be regulated in *trans* by miRNAs. One such miRNA is miR-71, which targets *alg-1* and acts in a negative feedback loop to control miRNA biogenesis/function (this work and Inukai et al.[21]). miR-71 has been shown to promote lifespan and stress response in *C. elegans* (this work and de Lencastre et al.[8]), and longevity of *glp-1* mutants depends on miR-71[9]. Although miR-71 is not increased when ALG-1 is overexpressed (i.e., CT20 worms), the levels of miR-71 are higher in *glp-1* mutants. This indicates upregulation of ALG-1 in *glp-1* mutants is not sufficient to increase miR-71, pointing to an additional mechanism to stabilize this miRNA in worms lacking the germline. Consistent with this hypothesis, there are twice as many miRNAs upregulated in *glp-1* mutants than in ALG-1/OE worms. There are multiple steps in miRNA biogenesis that can be subjected to regulation for specific miRNAs to be stabilized. One of them is through the regulation of pri-miRNA transcription. Indeed, we identified the transcription factor GEI-2 as a positive regulator of the *mir-71* promoter and found that *gei-2* loss-of-function requires miR-71 to induce ALG-1, possibly by alleviating the suppressive effect miR-71 exerts over ALG-1. However, *gei-2* is downregulated in *glp-1* mutants, and *gei-2* RNAi does not induce ALG-1 in these mutants. Hence, miR-71 levels are increased in *glp-1* mutants despite the reduction of *gei-2*. These findings support an alternative mechanism stabilizing miR-71 in *glp-1* mutants and reveal an intricate network controlling ALG-1 abundance at multiple levels. However, the level of induction of ALG-1 caused by the downregulation of *R02D3.7* and potentially other mechanisms is strong enough to overcome the upregulation of miR-71 (and potentially other negative regulators of ALG-1) in *glp-1* mutants. This mechanism allows the long-lived, stress-resistant *glp-1* mutant to induce miRNA biogenesis/function by upregulating ALG-1 despite maintaining high levels of miR-71[9,13].

Considering that upregulation of miR-71 is sufficient to promote longevity[9], the fact that miR-71 is not induced upon ALG-1 overexpression could also explain why ALG-1/OE worms are not long-lived under normal conditions (this work and Aalto et al.[17]). On the other hand, we observed that ALG-1 overexpression mimics the oxidative stress resistance phenotype of the germline-deficient mutant *glp-1*, one of the long-lived mutants that displays the highest level of ALG-1. Elevated levels of ALG-1 also delay larval development and reduce brood size. Similarly, worms with loss-of-function mutation in *alg-1* (allele *gk214*) exhibit various developmental impairments, including heterochronic defects and reduced progeny[16,58], while being short-lived[17,18]. These results underline the importance of ALG-1 and miRNAs in development and reproduction, and suggest that miRNA regulation may play a crucial role controlling the balance between reproduction and somatic maintenance[59].

ALG-1 overexpression only increases lifespan when worms are cultured in bacteria that do not produce NO. Considering that *C.*

*elegans* lacks nitric oxide synthase, the only source of NO worms are exposed to derive from the bacteria[32]. NO inhibits ALG-1 post-translationally via S-nitrosylation[30]. Hence, the absence of NO could further stimulate ALG-1 function. NO is also a free radical known to participate in redox mechanisms and reacts with superoxide[35,60]. Hence, in an NO-free environment, worms are expected to suffer from redox alterations, which include increases in superoxide levels and overreliance on superoxide dismutases to maintain redox balance, a condition that could sensitize worms to respond to ALG-1 overexpression.

In agreement, ALG-1 overexpression renders *C. elegans* more resistant to stress and this protection appears to be specific to oxidative stress, since ALG-1/OE does not protect worms from temperature stress and sensitizes them to reductive stress by DTT, further indicating that ALG-1 controls redox balance by pushing it towards a more reductive state. Our observations agree with a previous study by Finger et al. who show that ALG-1 is required for proper oxidative stress resistance in *C. elegans*[18]. On the other hand, in their model of *alg-1* loss-of-function, they observed decreased heat stress resistance and impaired proteostasis. Together, these results suggest that while *alg-1* deficiency reduces lifespan and increases vulnerability to multiple stressors, ALG-1 overexpression selectively improves survival under oxidative stress. Moreover, the complexity of the response and the intricate regulatory mechanism that ALG-1 is subjected to indicate that ALG-1 can be dynamically altered to confer oxidative stress protection, but should also be tightly regulated not to compromise other stress responses.

A plausible mechanism to explain how ALG-1 induction might confer protection against oxidative stress involves the upregulation of mitochondrial superoxide dismutase SOD-3. ALG-1 overexpression, like *alg-1* knockdown, induces the *sod-3* promoter. However, adding the antioxidant NAC to ALG-1/OE worms does not inhibit *sod-3* promoter activity, as it does in worms exposed to *alg-1* RNAi. This supports the notion that higher levels of ALG-1 preventively enhance *sod-3* promoter activity independently of endogenous pro-oxidant levels, aiding worms to endure oxidative stress. Consistent with these observations, *glp-1* mutants exhibit higher SOD-3 levels and have improved oxidative stress response (this study and[11,13]). Additionally, we demonstrated that the increased oxidative stress resistance of *glp-1* mutants is dependent on *alg-1*. Furthermore, while DAF-16 and SKN-1 contribute to the prolonged lifespan and stress resistance of *glp-1* mutants[13,39,41,53,61], the protective effects facilitated by ALG-1 overexpression also occur in conditions where *daf-16* and *skn-1* are silenced[11,13,39,41,53]. Together, these findings suggest that the upregulation of ALG-1 operates in tandem with DAF-16 and SKN-1, collectively enhancing stress resistance in germline-deficient mutants, in part manifested by upregulation of *sod-3*.

Contrasting previous data from the literature showing that *mir-71* loss-of-function mutants are more sensitive to oxidative stress[8], we did not see differences between the *mir-71(n4115)* null mutant and control worms under paraquat or sodium arsenite, although in our experiments the worms carried additional transgenes. Nevertheless, loss of *mir-71* blunted the oxidative stress protection promoted by ALG-1/OE. These results show that normal levels of miR-71 are required for stress resistance of worms overexpressing ALG-1. However, our model also contemplates the possibility that miRNAs other than miR-71 are playing a dynamic role in controlling oxidative stress response after upregulation of ALG-1. To identify these players, we measured miRNAs commonly upregulated in *glp-1* mutants and ALG-1/OE worms. Besides playing a fundamental role in the function of miRNAs, ALG-1 also participates in miRNA biogenesis by stabilizing the guide strand of miRNAs. In agreement, ALG-1 deficiency decreases global miRNA levels[15,16] and most of the miRNAs that fell within our cut-off ($\log_2$ fold change $> 1$ or $< -1$) were upregulated in ALG-1/OE worms and *glp-1* mutants. Of 11 miRNAs commonly upregulated in the two conditions, we showed that disruption of the *mir-87*, *mir-230*, and *mir-235* genes

sensitized worms to oxidative stress. Interestingly, these miRNAs are also upregulated in the dietary-restricted, long-lived mutant *eat-2*[62]. Moreover, miR-230-3p expression is increased in response to heat stress[63], while its expression decreases in aged worms[21]. miR-235-3p is also downregulated in aged worms, and mediates longevity induced by dietary restriction in *C. elegans*[64]. *mir-235* also has a protective effect on the response to exposure to graphene oxide[65], a toxic agent. These findings unveiled miRNAs induced by ALG-1 overexpression as relevant players in oxidative stress response.

Among the putative targets of these miRNAs, we found genes that are predicted to have their molecular function associated with PDI activity. PDIs are multifunctional enzymes with oxidoreductase, isomerase, and chaperone activities. PDI promotes the formation of disulfide bonds in newly synthesized ER proteins, primarily those destined to secretion or to the cell surface[66]. The thiol oxidase function of PDI is intricately linked to its reoxidation by ER oxidoreductin-1 (Ero1). This process involves the generation of reduced Ero1, which subsequently needs to be regenerated to its oxidized form. This regeneration occurs through electron transfer from reduced Ero1 to $O_2$, leading to the generation of hydrogen peroxide[66].

The ER lumen is known for its relatively oxidizing environment, characterized by high concentrations of hydrogen peroxide[67]. Concurrently, Ero1-dependent catalysis relies on the reduction of its non-catalytic regulatory disulfides, a process involving PDI or glutathione[68]. Thus, the loss of PDI may limit Ero1 activity, providing some level of protection against oxidative stress. However, this protective effect may come at the cost of impaired protein folding capacity, while simultaneously rendering cells more susceptible to reductive stress. Additional mechanisms associated with the effects of PDI depletion may include the loss of oxidized glutathione in the ER[68] and a diminished supportive interaction between PDI and oxidant-generating Nox NADPH oxidases[66]. These possible mechanisms remain to be investigated. On the other hand, general induction of miRNAs by upregulation of ALG-1 is anticipated to have a broad inhibitory effect on protein synthesis. Hence, ALG-1 upregulation and consequent downregulation of the PDI pathway may alter ER redox regulation and protein folding capacity through multiple mechanisms.

These observations concur with the fact that worms overexpressing ALG-1 are more sensitive to ER stress induced by the reducing agent DTT, while they cope better with exogenous exposure to pro-oxidant agents. In *C. elegans*, PDI-1, PDI-2, PDI-3, and PDI-6 are the best-characterized PDI proteins with homologies to human PDIs[69]. We found that miR-230-3p can directly target *pdi-2*. *C. elegans* mutants with loss-of-function of *pdi-2* exhibit severe morphological defects, uncoordinated movement, and adult sterility[70].

Here we used heterozygous mutants resulting in partial inhibition of *pdi-2* and we observed that limiting the enzyme improves oxidative stress resistance. Knockdown of *pdi-2* using RNAi results in similar protection from paraquat stress. Curiously though, *pdi-2* RNAi not only sensitizes worms to sodium arsenite, but also results in decreased lifespan under normal, unstressed conditions. These findings underscore the critical importance of fine-tuning the regulation of PDI-2. Taking into consideration the important role of PDI in maintaining redox balance in the ER while conferring proper protein homeostasis, it is expected that drastic changes in PDI-2 levels will lead to deleterious effects. Hence, whether *pdi-2* inhibition proves beneficial or detrimental may be a matter of the magnitude and the duration of the repression, as well as the nature of the stress.

Previous studies have indicated that neurons in *C. elegans* exhibit resistance to RNAi[71,72], whereas the *pdi-2* mutation is anticipated to impact all cells. Consequently, the varying outcomes observed with different loss-of-function models may also signify the influence of distinct tissues on the phenotypes. Given the tissue specificity of many miRNAs[73] and their established role in fine-tuning gene networks[5], we posit that miRNAs serve as cellular guardians of the PDI pathway,

preventing unbalanced changes that could lead to cellular dysfunction during stressful conditions, consequently influencing organismal stress resistance.

In conclusion, in the present work we described the mechanism through which the miRNA processing machinery is dynamically regulated to orchestrate organismal oxidative stress resistance under several conditions that lead to longevity, particularly in germline-deficient animals. Under a pro-oxidant environment (common in disease conditions), increasing a key enzyme that represents a limiting factor for miRNA function and biogenesis promotes survival of the organism. Importantly, increasing the efficiency of the miRNA machinery is not always beneficial, as it can also sensitize the organism to other kinds of stressors, decrease reproductive output, and delay development. Hence, it is not surprising that the physiological regulation of ALG-1 is tightly controlled by an intricate mechanism. Although some of the players participating in this mechanism as described here in *C. elegans* are not fully conserved, the tight regulation of components of the miRNA processing pathway and their role in stress response and aging have been clearly demonstrated in mammals[47,74–76]. In humans, germline mutations in argonautes (*AGO1* and *AGO2*) have been associated with defects in neurodevelopment[77,78], while somatic mutations in *AGO2* appear to be common in gastric and colorectal cancers with high microsatellite instability[79]. Future studies could reveal whether targeting these specific argonautes may serve to treat diseases associated with oxidative stress and improve human health.

## Methods

### Strains and maintenance of *C. elegans*
*C. elegans* were obtained from the Caenorhabditis Genetics Center (CGC). Worms were maintained in Petri dishes containing semi-solid Nematode Growth Media (NGM) supplemented with 1 mM $CaCl_2$, 1 mM $MgSO_4$, 25 mM $KPO_4$ and 5 μg/mL cholesterol, as described previously[80]. The NGM medium was supplemented with 100 μg/mL of streptomycin (Sigma-Aldrich, S6501-25G) for the maintenance plates seeded with OP50-1 (a streptomycin-resistant strain of OP50 *E. coli*), to avoid contamination. Bacteria were grown for 16 hours, during which they reached an optical density of approximately 1. Then, the culture was concentrated 10 times and inoculated onto plates containing NGM, forming a lawn above which the worms were cultivated. All *C. elegans* strains used are summarized in Supplementary Table 1. We used hermaphrodites in all experiments and males only for mating.

### RNAi
For RNAi experiments, NGM plates were supplemented with 1 mM $CaCl_2$, 1 mM $MgSO_4$, 25 mM $KPO_4$, 8 μg/mL cholesterol, 12.5 μg/mL tetracycline (Sigma-Aldrich, T7660), 100 μg/mL ampicillin (Sigma-Aldrich, A9518), and 1 mM Isopropyl-β-D-thiogalactopyranoside (IPTG) (Fisher Scientific, BP1755-1) to induce double-stranded RNA expression. *C. elegans* were fed with HT115 *E. coli* bacteria expressing the RNAi clones, which were obtained from the *C. elegans* ORFeome Library (Vidal RNAi library available through Horizon Discovery), kindly provided by Dr. Adam Antebi and Dr. Julio Ferreira. For control (ctrl) RNAi, HT115 expressing the L4440 (empty vector) clone was used.

### Lifespans assays
Approximately 120 synchronized day 0 (young adults) worms were transferred to lifespan plates containing *E. coli* OP50-1 (streptomycin-resistant strain) or *E. coli* HT115 (for RNAi experiments) bacteria and supplemented with 50 μg/mL 5-Fluoro-2'-deoxyuridine (FudR) (Thermo Scientific Chemicals, 227605000). Lifespans were performed at 20 °C under normal conditions. For lifespans under NO-free conditions, *B. subtilis* strain 1A1 (strain 168) and the isogenic *B. subtilis* ΔNOS, obtained from the Bacillus Genetic Stock Center (BGSC), were used. For lifespans under heat stress, a temperature of 30 °C was used. For oxidative stress induced by methyl viologen dichloride hydrate

(Paraquat) (Sigma-Aldrich, 856177), worms were transferred at L4 stage to lifespan plates supplemented with either 4 mM (for OP50-1 bacteria) or 8 mM of Paraquat (for HT115 bacteria). For reductive ER stress induced by DTT (dithiothreitol) (Thermo Fisher Scientific, D0632), the *C. elegans* strains were transferred at day 0 young adults to lifespan plates supplemented with 5 mM DTT. For all lifespans, dead worms were scored every 1 to 2 days from day 1, and animals that died by bagging, bursting, or crawling off the plates were censored.

### Oxidative stress induced by sodium arsenite
Using 96-well plates, day 3 adult worms were transferred to wells containing 100 μL of sodium arsenite (Sigma-Aldrich, S7400) solution in concentrations ranging from 10 to 15 mM, diluted in M9 buffer (22 mM $Na_2HPO_4$, 22 mM $KH_2PO_4$, 85 mM NaCl, 1 mM $MgSO_4$). After 8 h, dead worms were scored by touching their tail three times with a platinum wire. When touch response was absent, worms were considered dead. At least 6 wells containing 10 worms were used for each strain and each concentration. The experiments were repeated at least 2 times.

### Mobility analysis
Mobility was measured on days 0, 3, 6, and 9 of adulthood. Ten worms from lifespan plates were placed, individually, in a well of a 96-well plate containing 100 μL of M9 buffer and left at 20 °C for 10 min to acclimatize. The number of body bends per worm was measured for 30 s using a light stereoscope, and the values were multiplied by 2 to obtain the rate per minute. The measurements were repeated twice per worm and data represents the average.

### Pharyngeal pumping
Pharyngeal pumping analysis was done on days 0, 3, 6, and 9 of adulthood. Ten worms had their pharynx movements on lifespan plates observed using a light stereoscope. The number of pharynx movements per worm was measured for 30 s, and the values were multiplied by 2 to obtain the rate per minute. The measurements were repeated twice per worm and data represents the average.

### Duration of development
Individual adult worms were allowed to lay eggs for one hour on OP50-1 plates. After 60 hours of progeny development, the plates were monitored every hour using a light stereoscope to score for adults, identified by the presence of eggs in the uterus. This monitoring continued until the last worm reached the adult stage.

### Brood size assay
Approximately 200 eggs laid by adult worms were transferred to OP50-1 plates and maintained until day 0 (young adults). Each adult worm was then individually transferred to new plates and maintained for 24 hours. After 24 hours, the worms were transferred to fresh wells for another 24-hour incubation, while the eggs/larvae from the previous well were counted. This procedure was repeated until day 5 of adulthood.

### ModENCODE analysis
To look for candidates for *alg-1* transcriptional regulators, version 33 of the modENCODE ChIP-Sequencing consortium[24] was used. The candidates were picked by selecting genes that had a significant peak in the region of up to 2000 base pairs upstream of *alg-1* initiation codon (X chromosome: 13947420..13949420).

### GFP reporter analyses
For measurements of the *alg-1p::alg-1::gfp* reporter (*ap423* allele obtained from the PQ530 strain) in the long-lived mutants, the strains were maintained on OP50-1, and had their fluorescence measured on day 1 of adulthood. Ten to twenty worms were immobilized using 0.1% sodium azide (Sigma-Aldrich, 71290) on M9 solution and placed on

glass slides. The images were acquired using the ZEISS Apotome 3 (Zeiss microscopy). For experiments using the other reporters, the strains were transferred to plates containing 50 µg/mL FudR (Thermo Scientific Chemicals, 227605000) on day 0 of adulthood. For *sod-3* and *gst-4* analyses, plates were also supplemented with 4 mM Paraquat (Sigma-Aldrich, 856177), 10 mM N-acetylcysteine (NAC) (Sigma-Aldrich, A7250) or vehicle (M9) solution. On day 3 of adulthood, ten adult worms were transferred to transparent 96-well plates containing 100 µL of 0.1% sodium azide (Sigma-Aldrich, 71290) in M9 buffer, for immobilization, and had their images acquired using Cytation™ 5 Cell Imaging Multi-Mode Reader (BioTek Instruments). For all measurements using RNAi, worms were grown on ctrl RNAi (empty vector) plates until day 0 and then transferred to the experimental RNAi until the day fluorescence was measured. The software ImageJ Fiji (National Institute of Health) was used for image analysis. The mean fluorescence intensity of each worm was calculated, and the normalization was performed by subtracting the mean intensity values of worms without fluorescence (WT) and dividing these values by the mean intensity values of the control group.

## Real-time qPCR

RNA from 300 day-1 adult worms was extracted using TRIzol Reagent (Thermo Fisher Scientific, 15596018). Reverse transcription was performed using 200 ng of total RNA and High-Capacity cDNA Reverse Transcription kit (Thermo Fisher Scientific, 4368814) according to the manufacturer's protocol. Real-time PCR was conducted using SYBR Green Master Mix (Thermo Fisher Scientific, 4309155) and primers at the final concentration of 0.5 µM. Fluorescence was detected using the CFX384 Detection System (Bio-Rad Laboratories). Real-Time PCR System following the protocol: 95 °C−5 min and 40 cycles of 95 °C −15 s, 60 °C−20 s, and 72 °C−30 s. Expression levels were normalized using the reference gene *Y45F10D.4*. Primers used: *alg-1* forward - CGCGCTCGTTAATCATCTTGT, *alg-1* reverse - GGATGAACCTGCACAGCTC, *Y45F10.4* forward - CGAGAACCCGCGAAATGTCGGA, *Y45F10.4* reverse - CGGTTGCCAGGGAAGATGAGGC.

## Small RNA sequencing

Total RNA was extracted from pools of 500–800 *C. elegans* at day 0 using TRIzol reagent and following the manufacturer's instructions (Thermo Fisher Scientific, 15596018). Samples of 6 µg of total RNA were sent for small RNA sequencing at the Genomics Center of ESALQ-USP using the HiSeq2500 platform (Illumina). After preprocessing the data with the Fastx-ToolKit software to remove adapters and select reads with a minimum length of 18 base pairs (bp), the analysis was conducted using the Bowtie2 software with the ce10 reference genome provided by UCSC. The data was normalized by the total number of reads generated per sample. miRNAs with $\log_2$ fold change > 1 or < −1 when comparing to their control groups [N2 for *glp-1(e2144)* and ctrl for ALG-1/OE] were considered differentially expressed and were plotted in a heatmap generated using GraphPad Prism (version 8.0). Venn diagrams were created using the following website: http://bioinformatics.psb.ugent.be/webtools/Venn/. The complete raw data can be accessed through the Gene Expression Omnibus (GEO) by the accession number GSE260938.

## RNAseq data analysis

RNAseq data (GSE111338) was assessed at GEO and used to retrieve gene expression data of *glp-1(e2141)* vs. WT worms at day 1. For *eat-2(ad1116)*, *raga-1(ok386)*, *clk-1(qm30)*, and *age-1(hx546)* mutants at day 1, we used the RNAseq data from Dutta et al.[23], available upon request from the Mair lab. Statistical analysis was performed using EdgeR package, on RStudio (version 1.3.1). The data were normalized with "calcnormFactors()" using the "glmQLFTest()" method. Data were considered statistically different when *P* value and false discovery rate (FDR) were less than 0.05.

## Gene ontology of miRNA targets

The TargetScanWorm Release 6.2[44] was used to assess predicted miRNA targets. Predicted targets with aggregate Pct > 0.19 were selected and intersected with genes upregulated on a microarray of *alg-1(gk214)* worms[45] (GSE19138) curated by WormExp v2.0[81]. The list of intersected genes had their gene ontology analyzed on WormEnrichr[46], using the gene set library GO molecular function AutoRIF Predicted z-score.

## pdi-2 3'UTR cloning

To clone the putative binding site of miR-230-3p located at the *pdi-2* 3'UTR into the 3'UTR of a luciferase reporter vector, oligonucleotides (*pdi-2* sense CTAGTTAGCGGCCGCTAGTATCATCTCATCTT TACTAATAAA; *pdi-2* antisense AGCTTTccggtcTAAAGATGAccTGAT ACTAGCGGCCGCTAA; mutated *pdi-2* sense CTAGTTAGCGGCCG TAGTATCAggTCATCTTTAgaccggAA; mutated *pdi-2* antisense AG CTTTccggtcTAAAGATGAccTGATACTAGCGGCCGCTAA) were synthesized, diluted to a concentration of 1 µg/µL and annealed in annealing buffer (10 mM Tris, pH 7.5 – 8.0; 50 mM NaCl; 1 mM EDTA). The solution was heated at 95 °C for 2 min, allowed to cool to room temperature and kept in this condition for 45 min. 1 µg of pMIR-REPORT™ vector (Thermo Fisher Scientific, AM5795) was digested with 1 µL (10 units) of restriction enzymes *SpeI* (New England Biolabs, NEB #R3133) and *HindIII* (New England Biolabs, NEB #R3104), in the presence of Cutsmart buffer (New England Biolabs, NEB #B6004), to a final volume of 50 µL. The annealed oligos were diluted to 4 ng/µL and ligated to 50 ng of pMIR-REPORT, in the presence of T4 DNA ligase (New England Biolabs, M0202S) and 1x ligase buffer (New England Biolabs, B0535S) in a final volume of 20 µL. 2 µl of the ligation was transformed into competent DH5α bacteria according to the Addgene protocol (https://www.addgene. org/protocols/bacterial-transformation) and then 100 µL of the transformation were plated on LB-agar plates containing 100 mg/ mL of ampicillin (Sigma-Aldrich, A9518-25g). Clones were selected and plasmids were isolated using the QIAprep Spin miniprep kit (Qiagen, 27104).

## miR-230-3p target assay

HEK293T cells were seeded into 96-well plates and, after 24 hours, they were transfected with plasmids carrying the luciferase gene under the control of the native or mutated *pdi-2* 3'UTR sequences (1 µg/mL) in the presence of the *C. elegans* miR-230-3p (Thermo Scientific, 4464066) or miR-85-3p (negative control) (Thermo Scientific, 4464066) mimics at the concentration of 17.5 nM. The vector pRL-SV40 (which expresses Renilla luciferase) was co-transfected for normalization. Plasmids were transfected using the Effectene Transfection Reagent kit (Qiagen, 301425) and miRNA mimics using the HiPerFect Transfection Reagent kit (Qiagen, 301704). After 24 hours of transfection medium was exchanged, and 24 hours later the cells were washed with PBS and luminescence of Firefly Luciferase and Renilla Luciferase was measured using the Dual-Glo® Luciferase Assay kit (Promega, E2920).

## Statistics

Lifespan assays were compared using the log-rank test. Two-tailed Student's t-test was used to compare two means, and one-way ANOVA was used to compare more than two means. In experiments with two independent variables, means were compared using two-way ANOVA. We assumed normal distribution for statistical comparisons. GraphPad Prism (version 8.0) was used to plot data and to calculate statistics. $P < 0.05$ was considered statistically significant.

## Reporting summary

Further information on research design is available in the Nature Portfolio Reporting Summary linked to this article.

## Data availability

All data generated or analyzed during this study are provided in the Supplementary Information and Source Data files and are available in public repositories or upon request from their original sources. The small RNA sequencing data generated in this study (Fig. 4a–e) have been deposited in the GEO database under accession code GSE260938. The processed data is also available in the Supplementary Data 3. The RNA sequencing data re-used in this study (Fig. 1a, g and Supplementary Fig. 2a) are available in the GEO database under accession code GSE111338 or were shared by and is available upon request from the Mair lab[23]. The processed data used in the study are also available in the Supplementary Data 1. The microarray data re-used in this study (Fig. 5a and Supplementary Fig. 9) are available in the GEO database under accession code GSE19138. The processed data used in this study is also available in the Supplementary Data 4. Source data are provided with this paper.

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

## Acknowledgements

We thank Elzira E. Saviani and Anne Lanjuin for technical support, and the *C. elegans* Genetics Center (University of Minnesota) – funded by the NIH Office of Research Infrastructure Programs (P40 OD010440) – for providing *C. elegans* strains. We also thank the Genomics Center of

ESALQ-USP for the small RNA sequencing, and Adam Antebi (Max Planck Institute for Biology of Ageing) and Julio Ferreira (University of São Paulo) for RNAi clones. We thank Francisco Laurindo (University of São Paulo) for reagents and discussion. This work was funded by the Conselho Nacional de Desenvolvimento Científico e Tecnológico (CNPq) – grant number 310287/2018-9 (M.A.M.), Coordenação de Aperfeiçoamento de Pessoal de Nível Superior – Brasil (CAPES) – Finance Code 001, 88887.340135/2019-00 (C.A.V.J), 88887.489628/2020-00 (G.T.S) 88881.143924/2017-01 (M.A.M.), Fundação de Amparo à Pesquisa do Estado de São Paulo (FAPESP) – grant numbers: 2019/01587-1 (C.A.V.J), 2016/15958-3 (R.P.M.), 2017/04377-2 (S.P.), 2023/08316-9 (E.A.S.), 2022/05851-8 (E.A.S.), 2014/10814-8 (E.A.S.), 2017/01339-2 (H.C.), 2018/11792-9 (D.L.B), 2019/04726-2 (T.L.K), 2018/21635-8 (G.P.R.), 2017/23920-9 (R.G.L.), 2014/50897-0 (K.B.M.), 2021/08354-2 (M.A.M.), 2019/25958-9 (M.A.M.), 2017/01184-9 (M.A.M.), and National Institutes on Aging - R01AG044346, R01AG059595 (W.B.M).

## Author contributions

C.A.V.J.: Conceptualization, Formal analysis, Investigation, Writing – Original Draft, Visualization, Project administration; R.P.M.: Conceptualization, Formal analysis, Investigation, Project administration, Review & Editing; S.P.: Investigation, Validation, Review & Editing; E.A.S.: Investigation, Validation, Review & Editing; H.C.: Conceptualization, Investigation, Review & Editing; D.L.B.: Investigation, Validation; G.T.S.: Investigation, Validation, Review & Editing; T.L.K.: Investigation, Validation; G.P.R.: Investigation, Validation; R.G.L.: Investigation, Validation; K.B.M.: Resources provision, Funding acquisition, Review & Editing; W.B.M.: Conceptualization, Review & Editing, Resources provision; M.A.M.: Conceptualization, Writing – Review & Editing, Project administration, Funding acquisition, Resources provision.

## Competing interests

The authors declare no competing interests.
