## [Peer Review File · Nature Communications]

An Intricate Network Involving the Argonaute ALG-1 Modulates Organismal Resistance to Oxidative StressReviewer #1 (Remarks to the Author):

In this interesting manuscript, the authors analyse significantly extend the role of argonaute ALG-1 (Ago-like gene 1) in *C. elegans* in regards to resilience against both oxidative stress and heat.

(i) For which reason(s) do the authors focus on ALG-1 only? I.a.w. what is the rationale to (at least initially) omit ALG-2? I believe some pre-existing (see (ii)) and/or novel experimental evidence would be helpful.

(ii) The authors should definitely cite a previous paper by the Hoppe lab (PubMedID 34723149, 2021) entitled 'The Argonaute Proteins ALG-1 and ALG-2 Are Linked to Stress Resistance and Proteostasis'.

There, it has been shown that mostly ALG-1, and to a lesser extent also ALG-2, are required for resistance against oxidative stress; moreover, the authors show there that ALG-1 and ALG-2 are both independently required for resistance against heat stress. Both findings are consistent with the present manuscript where ALG-1 only has been studied.

(iii) Fig. 1D and E: If activation of *sod3* and *gst4* promoters is increased in states of *agl-1*-knockdown, how would this explain the increase in paraquat sensitivity? Shouldn't these worms endure paraquat better than control RNAi nematodes?

(iv) Fig. 3: Are developmental times decreased in *alg-1* overexpressors (in analogy to Fig. 1A)?

(v) Does *alg-1* RNAi impact the lifespan of long-lived *glp-1* mutants?

(vi) What is the total and time-resolved egg laying capacity of N2 and *glp-1* mutants in a) states of *alg-1* RNAi application, and b) states of *alg-1* overexpression?

(vii) Fig. 4: Does R02D3.7 RNAi increases lifespan in N2 in the absence of arsenite?

(viii) Fig. 4: Does R02D3.7 RNAi increases lifespan in N2 in the presence of paraquat?

(ix) Figs. 4 and 7: Does R02D3.7 RNAi impacts lifespan of *alg-1* mutants as well as *alg-1* overexpressors?

(x) Fig. 5: Do *mir-87(n4104)*, *mir-230(n4535)* or *mir-235(n4504)* mutants impact the increase in longevity of *alg-1* overexpressors?

(xi) Fig. 6: Do chemical inhibitors of PDIs impact lifespan of N2 per se, as well as on paraquat and arsenite?

(xiii) Fig. 6: Do chemical inhibitors of PDIs impact lifespan of *alg-1* overexpressors?

(xiv) Fig. 6: Since the PDI pathway is closely linked to ER function and proteostasis, the above-mentioned reference (PubMedID 34723149, 2021) is particularly relevant to mention here. The paper anticipates the current findings to a certain while limited extent.

(xv) Fig. 7: Does *alg-1* overexpression compensate for the reduced longevity of the *skn-1 zu135* strain?

Reviewer #2 (Remarks to the Author):

In this manuscript Vergani et al. dissect the role of ALG-1, the major microRNA Argonaute in *C. elegans*, in stress resistance and lifespan. Prior work established that *alg-1* promotes long lifespan upstream of the *C. elegans* FOXO protein DAF-16. Similar to *daf-16* mutants, the authors found that *alg-1* mutants were sensitive to oxidative stress and expressed markers associated with oxidative stress. Next, the authors used published mRNA-seq data as well as a translational

reporter to establish that *alg-1* is overexpressed in four different long-lived mutant backgrounds, most prominently in the germline-less *glp-1* mutant. Starting with modENCODE ChIP-seq data, the authors identified five transcription factors that bind the *alg-1* promoter and negatively regulate the expression of a GFP::*alg-1* transgene. One of these transcription factors, R02D3.7, is the best candidate of a direct regulator of *alg-1* expression because it is downregulated in all of the lifespan mutants where *alg-1* is upregulated. They provide evidence that a second transcription factor, GEI-1, regulates *alg-1* expression indirectly via *mir-71*. To complement the loss-of-function experiments, the authors overexpress *alg-1* and show increased resistance to oxidative stress and upregulation of one oxidative stress marker. The use of two different methods to overexpress ALG-1 greatly strengthens their conclusions, and the transgene overexpression includes a very nice control of a similar transgene that does not express *alg-1*. Consistent with R02D3.7 as a negative regulator of *alg-1*, RNAi of R02D3.7 increased resistance to oxidative stress in an *alg-1*-dependent manner. The authors characterized changes in miRNA levels in *glp-1* mutants and in *alg-1* overexpressing strains, thereby identifying three microRNA mutants that are sensitive to oxidative stress: *mir-87*, *mir-230*, and *mir-235*. Finally, they identified a predicted miR-230 target, *pdi-2*, as a negative regulator of stress resistance.

This paper is interesting, well-written, and thoughtful. The role of *alg-1* and microRNAs in stress resistance has not previously been explored and this paper provides new insight from the regulation of *alg-1* all the way to a particular microRNA target. The data are clear and convincing. The comments below are intended to further strengthen this work. All of the changes I suggest could be addressed in the writing. However, if the authors wish to provide more evidence for their claims, I also mention some examples of possible experiments that could be carried out.

Major comments:

1. The details of transgenes and CRISPR mutants are not clear to me from the methods, Supplemental Table, or text of the paper. It is difficult for me to interpret findings from these tools without an understanding of what specific sequences are present in these strains. If the transgenes and CRISPR mutants are derived from published work, please cite the relevant papers. If any of them were constructed for this paper, please provide the details in the Methods section.
2. The experiment in Fig 6B showed that miR-230 can target the *pdi-2* 3' UTR in mammalian cell culture, but it does not show that *pdi-2* is a true target of miR-230 in *C. elegans*. The authors can either adjust their language accordingly or perform the experiment in *C. elegans*, for example by tagging the endogenous *pdi-2* locus and examining expression in a miR-230 mutant vs. wild-type. If desired, the argument could be further strengthened by mutating the miR-230 sites in the *pdi-2* 3' UTR.
3. *daf-16* is well characterized to promote resistance to oxidative stress and the reason the authors provide for investigating the role of *alg-1* in oxidative stress is the known regulation of *daf-16* by *alg-1*. However, the interplay of *daf-16* and the new factors characterized here is not considered. The authors provide arguments in the discussion (lines 369-387) that *alg-1* can regulate stress resistance and longevity independently of *daf-16* and this argument seems reasonable as a general statement, but no data are provided to specifically address this point. The authors could either present possibilities in the discussion or perform experiments, for example testing the ability of reducing *pdi-2* function to suppress oxidative stress sensitivity of *daf-16* mutants or examining the expression of miR-230 and *pdi-1* in *alg-1* mutants vs. *daf-16* mutants.
4. From what I could see in Supplemental Table 6, it appears there are two different GFP::*alg-1* reporters: One I assume from the "Is" name to be an exogenous multicopy transgene with the *alg-1* promoter, GFP fused to the N-terminus of ALG-1, and the *alg-1* 3'UTR. The other reporter is the endogenous *alg-1* allele tagged with GFP, also at the N-terminus. Why was the screen for transcription factors that regulate *alg-1* expression done with the exogenous transgene? The findings would appear to be more meaningful in the endogenous context. If the authors want to strengthen the arguments in Fig 2 and Fig S1, key experiments could be repeated in the endogenous context.

Minor points:

1. It would be helpful to specify that the "ctrl" RNAi is empty vector in the figure legends, so the reader does not need to search the methods and know what L4440 is to understand the experiment.

2. Please explain the term "OP50-1", as opposed to the more commonly used "OP50" bacteria.

3. It is interesting that both alg-1 RNAi and overexpression of alg-1 cause increased expression of sod-3p::GFP (Fig. 1D & Fig 3E). Consider including a brief comment explaining why this may be the case.

Response to the Reviewers

We thank the Reviewers for dedicating valuable time to thoroughly review our manuscript and for providing insightful feedback. The positive remarks, particularly noting the manuscript's "well-written" and "thoughtful" nature, as well as the clarity and persuasiveness of the data, are truly appreciated. We also acknowledge and thank the Reviewers for identifying areas that warranted editing, clarification, or additional data. Their constructive comments have significantly contributed to enhancing the quality of our work. We have addressed each one of the Reviewers' comments, incorporating necessary revisions into this updated version of the manuscript. We believe that the additional experiments and textual modifications not only address the Reviewers' points but also contribute to a more robust and coherent presentation of the phenomenon under study, leading to an improved manuscript.

Enclosed is a point-by-point response (in blue) outlining how we addressed each of the Reviewers' comments. Additionally, we have included a version of the revised manuscript with all changes highlighted to facilitate understanding of the modifications made.

Reviewer #1:

In this interesting manuscript, the authors analyse significantly extend the role of argonaute ALG-1 (Ago-like gene 1) in *C. elegans* in regards to resilience against both oxidative stress and heat.

R: We express our gratitude for the Reviewer's comments and acknowledgement of our work. Find below a point-by-point response to these comments.

(i) For which reason(s) do the authors focus on ALG-1 only? I.a.w. what is the rationale to (at least initially) omit ALG-2? I believe some pre-existing (se (ii)) and/or novel experimental evidence would be helpful.

R: We thank the Reviewer for bringing attention to this reasoning, allowing us to provide a rationale and additional experimental evidence supporting our decision to concentrate on ALG-1, as opposed to ALG-2. While both ALG-1 and ALG-2 contribute to miRNA function and biogenesis in *C. elegans*, their roles in lifespan regulation and stress resistance exhibit notable distinctions. Notably, the literature indicates varying outcomes for *alg-2* loss-of-function mutants in terms of longevity, ranging from long-lived to short-lived in different studies^{1,2}. In contrast, *alg-1* loss-of-function mutants consistently demonstrate reductions in lifespan and stress resistance, as confirmed by multiple groups. Furthermore, *alg-1* loss-of-function mutants display higher sensitivity to oxidative stress compared to *alg-2* loss-of-function mutants². Importantly, the revised manuscript incorporates additional evidence indicating that, in various long-lived strains, *alg-1* levels are induced, whereas the levels of *alg-2* do not consistently correlate with longevity (see below and in new Supplementary Fig. 2a). These aspects support our decision to narrow our focus to ALG-1 regulation, given its more consistent and pivotal role in *C. elegans* lifespan and stress resistance. This rationale and discussion, along with the new data, have been integrated into the revised version of the manuscript

Figure R1. *alg-1* and *alg-2* mRNA expression in long-lived mutants as determined by RNAseq.

(ii) The authors should definitely cite a previous paper by the Hoppe lab (PubMedID 34723149, 2021) entitled 'The Argonaute Proteins ALG-1 and ALG-2 Are Linked to Stress Resistance and Proteostasis'.

There, it has been shown that mostly ALG-1, and to a lesser extent also ALG-2, are required for resistance against oxidative stress; moreover, the authors show there that ALG-1 and ALG-2 are both independently required for resistance against heat stress. Both findings are consistent with the present manuscript where ALG-1 only has been studied.

R: We thank the Reviewer for highlighting this oversight. We would like to apologize for this omission, as we fully concur with the Reviewer regarding the importance of citing the referenced study in our manuscript. Although we had previously discussed the study internally within our research team and acknowledge its alignment with our study, we regrettably overlooked its inclusion during the manuscript writing process. Recognizing its significance, we have now rectified this oversight in the revised version of the manuscript (Reference 18). In fact, we have incorporated multiple citations within the text to underscore its relevance not only in acknowledging prior evidence that supports our model but also as a justification for our emphasis on ALG-1, as elaborated in the response to the preceding point.

(iii) Fig. 1D and E: If activation of *sod3* and *gst4* promoters is increased in states of *agl-1*-knockdown, how would this explain the increase in paraquat sensitivity? Shouldn't these worms endure paraquat better than control RNAi nematodes?

R: The Reviewer raises a good point. To explain this, we hypothesized that the increases in *sod-3* and *gst-4* promoter activities could reflect a persistent state of oxidative stress induced in worms subjected to *alg-1* RNAi. According to this scenario, one would expect worms where *alg-1* is knocked down to struggle to manage an additional influx of pro-oxidants effectively, as we observe. Further supporting this hypothesis, our data show that both paraquat exposure and *alg-1* RNAi increase *sod-3* and *gst-4* promoter activity to a comparable extent, with no additive effects observed when paraquat is added to *alg-1* RNAi (see Supplementary Fig. 1c and d). Nonetheless, to directly test this hypothesis, we conducted a new experiment employing the antioxidant N-acetylcysteine (NAC). NAC effectively suppressed the elevated *sod-3* promoter activity in worms exposed to *alg-1* RNAi, but not in worms overexpressing *alg-1* (refer to new Fig. 2i and below). Moreover, in worms overexpressing *alg-1*, paraquat further increased *sod-3* promoter activity levels (see Fig. 2g). These findings support the notion that the activation of *sod-3* promoter in *alg-1* deficient worms is a consequence of endogenously produced pro-oxidants, indicative of an inherent state of oxidative stress. Conversely, in worms overexpressing *alg-1*, the upregulation of the *sod-3* promoter occurs independently of the redox state, or at least remains insensitive to pro-oxidants mitigated by NAC. This suggests a direct regulatory mechanism that protects worms from oxidative stress upon ALG-1 overexpression. The manuscript has been updated to incorporate these new data, accompanied by a detailed discussion of the results.

Figure R2. *sod-3* promoter activity upon *alg-1* RNAi or overexpression in worms treated with N-acetylcysteine (NAC) or vehicle.

(iv) Fig. 3: Are developmental times decreased in *alg-1* overexpressors (in analogy to Fig. 1A)?

R: In response to the Reviewer's suggestion, we conducted an experiment to assess the developmental time in *alg-1* overexpressors. The results, depicted below and presented in the new Supplementary Fig. 4g of the manuscript, reveal a significant delay in development among worms overexpressing *alg-1*. This observation aligns with the concept that precise regulation of miRNA levels is essential for ensuring proper developmental processes. The data was included in the manuscript along with this discussion.

Figure R3. ALG-1 overexpression delays development in *C. elegans*.

(v) Does *alg-1* RNAi impact the lifespan of long-lived *glp-1* mutants?

R: In response to the Reviewer's suggestion, we conducted an experiment to explore the role of ALG-1 in lifespan extension in *glp-1* mutants. While ALG-1 overexpression is not sufficient to impact lifespan under normal circumstances (i.e., without stressors) (Supplementary Fig. 4a and b), we recognize the significance of investigating its involvement in longevity induced by

germline deficiency, a condition where ALG-1 is upregulated. The results, represented below and depicted in the new Fig. 2f added to the manuscript, demonstrate that the extended lifespan observed in *glp-1* mutants is contingent upon the presence of ALG-1. This evidence aligns with our proposed model, highlighting ALG-1 as a pivotal mediator in the mechanisms governing longevity in response to germline deficiency.

Figure R4. Longevity of *glp-1* mutants requires ALG-1.

(vi) What is the total and time-resolved egg laying capacity of N2 and *glp-1* mutants in a) states of *alg-1* RNAi application, and b) states of *alg-1* overexpression?

R: In response to the Reviewer's recommendation, we examined the reproductive capacity of worms with altered ALG-1 expression levels, either through overexpression or knockdown. Our findings revealed a consistent reduction in brood size in both scenarios (see results below). When applying *alg-1* RNAi to the ALG-1 overexpressing strain, the effects were neither additive nor suppressed, supporting an expected interaction between the interventions and suggesting a dominant effect of one of them (likely the RNAi). More importantly, these results show that reproduction is highly sensitive to ALG-1 levels, aligning with the established role of miRNAs in this biological process. Brood size measurements were not conducted with *glp-1* mutants due to their sterility at restrictive temperatures, as they lack the germline.

Figure R5. Brood size in worms where *alg-1* was silenced (RNAi) and/or overexpressed (OE).

In incorporating these results into the revised manuscript, along with additional data generated for this revision, we aimed to maintain a focus on the foundational and most novel aspect of the paper—namely, the physiological upregulation of ALG-1 and its impact on oxidative stress resistance. To streamline this presentation, in the revised manuscript we have chosen to emphasize data from the ALG-1 overexpression model. Hence, the figure containing *alg-1* RNAi data has been relocated to the supplementary material, in which we displayed the phenotypes associated with longevity and oxidative stress resistance. That considered, all relevant data pertaining to the role of ALG-1 in stress resistance (gain- or loss-of-function), health span and longevity has been incorporated into the manuscript, along with evidence demonstrating the impact of ALG-1 overexpression on development and reproduction (new Supplementary Fig. 4g and h), as requested by the Reviewer.

(vii) Fig. 4: Does R02D3.7 RNAi increases lifespan in N2 in the absence of arsenite?

(viii) Fig. 4: Does R02D3.7 RNAi increases lifespan in N2 in the presence of paraquat?

(ix) Figs. 4 and 7: Does R02D3.7 RNAi impacts lifespan of *alg-1* mutants as well as *alg-1* overexpressors?

R: We acknowledge the valuable suggestions provided by the Reviewer to delve deeper into the role of R02D3.7 in longevity and stress resistance. In the prior version of the manuscript, we had identified R02D3.7 as a negative regulator of *alg-1*, establishing its control over oxidative stress resistance induced by arsenite in an *alg-1* dependent manner. Recognizing the opportunity for a more thorough characterization of R02D3.7 as a stress-related transcription factor, we used this chance to generate more data and learn more about the gene, given the limited existing knowledge about R02D3.7 in the literature. We then conducted lifespan measurements on

worms subjected to R02D3.7 RNAi and observed no discernible changes (refer to the data below and in the new Fig. 3f of the revised manuscript). Furthermore, R02D3.7 RNAi did not impact the reduced lifespan of worms subjected to *alg-1* RNAi (refer to the data below and in the new Fig. 3f of the revised manuscript), nor did it sensitize ALG-1 overexpressing worms to an extended lifespan (refer to data below, only to the Reviewer). These results demonstrate that, like ALG-1 overexpression, R02D3.7 loss-of-function does not alter the lifespan of *C. elegans* under normal conditions.

Figure R6. Lifespan of wild type control (WT - ctrl) or *alg-1* overexpressor (ALG-1/OE) worms upon *alg-1* and/or R02D3.7 RNAi.

In contrast, under paraquat exposure, R02D3.7 RNAi significantly increased worm survival (refer to data below and in the new Fig. 3d of the revised manuscript). R02D3.7 RNAi could still increase protection from paraquat stress even when *alg-1* was knocked down, whereas the addition of *alg-1* RNAi decreased paraquat resistance when R02D3.7 was silenced. These results suggest the effect of R02D3.7 RNAi extends beyond its effect over *alg-1*. Furthermore, in contrast to the observed effects on WT worms, the knockdown of R02D3.7 showed limited impact on ALG-1/OE worms, failing to further increase resistance to paraquat (refer to data below and in the new Fig. 3e of the revised manuscript). Collectively, these findings lend further support to our proposed model, demonstrating that while the downregulation of R02D3.7, which results in elevated ALG-1 levels, is insufficient to extend lifespan, it does confer protection against oxidative stress, and this mechanism operates, at least partially, via ALG-1, and can be mimicked by ALG-1 overexpression.

Figure R7. Survival on paraquat of wild type control (WT – ctrl) or *alg-1* overexpressor (ALG-1/OE) worms upon *alg-1* and/or R02D3.7 RNAi.

(x) Fig. 5: Do *mir-87*(n4104), *mir-230*(n4535) or *mir-235*(n4504) mutants impact the increase in longevity of *alg-1* overexpressors?

R: We conducted the experiments in line with the Reviewer's suggestion and found that, akin to worms with normal ALG-1 levels, the *mir-87*(n4104), *mir-230*(n4535), and *mir-235*(n4504) mutations diminish the survival of *alg-1* overexpressors under paraquat stress (refer to the data below and in the new Supplementary Fig. 8 of the revised manuscript). Regarding arsenite stress, only the *mir-235*(n4504) mutation demonstrated a reduction in the survival of *alg-1* overexpressors, mirroring the findings in wild-type worms (refer to the data below and in the new Supplementary Fig. 8b of the revised manuscript). These results underscore the pivotal role of these specific miRNAs in mediating the stress resistance conferred by ALG-1 upregulation. We did not investigate the role of these miRNAs in longevity of *alg-1* overexpressors as these worms are not long-lived under unstressed conditions. Thus, we limited our experiments to oxidative stress conditions where ALG-1 overexpression is protective.

Figure R8. Survival on paraquat or arsenite stress of *alg-1* overexpressor (ALG-1/OE) worms upon *mir-87*, *mir-230* or *mir-235* deficiency. *Mir-87*(n4104), *mir-230*(n4535) and *mir-235*(n4504) are null alleles.

(xi) Fig. 6: Do chemical inhibitors of PDIs impact lifespan of N2 per se, as well as on paraquat and arsenite?

(xiii) Fig. 6: Do chemical inhibitors of PDIs impact lifespan of alg-1 overexpressors?

R: We appreciate the constructive suggestions from the Reviewer regarding the use of chemical inhibitors of PDIs to further substantiate the involvement of these enzymes in longevity and stress resistance. Employing bepristat 1a, a known selective reversible inhibitor of PDI³, we initially aimed to assess its functionality in *C. elegans*. To this end, we exposed a *hsp-4p::GFP* reporter strain to the inhibitor at a concentration 15 μ M (higher than half-maximal inhibitory concentration (IC50) of $\sim 0.7 \mu$ M³) to inhibit PDI and measured GFP fluorescence as an indicator of ER stress. Given the role of PDI in ER stress, we expected the *hsp-4* promoter activity to be induced upon PDI pharmacological inhibition. In fact, silencing *pdi-2* with RNAi resulted in an approximately 7-fold increase in GFP fluorescence. In contrast, bepristat did not increase (in fact, reduced) GFP levels in the *hsp-4p::GFP* reporter strain. Although these results were unexpected and might suggest that bepristat does not inhibit PDI activity in *C. elegans*, we proceeded with longevity and oxidative stress assays using bepristat, comparing its effects with *pdi-2* RNAi as a reference.

In summary, our results demonstrate that bepristat does not influence lifespan or oxidative stress resistance, irrespective of whether worms overexpressed ALG-1 or not (refer to the data below, only to the Reviewers). In stark contrast, *pdi-2* RNAi enhanced paraquat resistance, diminished survival under arsenite exposure, and reduced worm lifespan (refer to the data below and new Fig. 5). The divergent results between the chemical and genetic approaches could be attributed to several possibilities. Although bepristat has been demonstrated to inhibit PDI activity by blocking substrate binding, it paradoxically enhances PDI catalytic activity in certain contexts³. The extent to which this mechanism occurs in worms, and whether PDI function in *C. elegans*, controlling stress resistance, operates in a substrate-independent catalytic manner that is insensitive to bepristat, remains an open question.

Considering these uncertainties, we opted not to include the experiments with bepristat in the manuscript. However, we did include the RNAi experiments as additional evidence that partial loss of *pdi-2* can protect worms from paraquat, while resulting in increased sensitivity to arsenite stress and reducing lifespan. Indeed, these RNAi experiments unveiled a crucial finding—namely, the necessity for fine-tuning the levels of *pdi-2* for optimal efficacy in conferring oxidative stress resistance. If the levels of *pdi-2* are either too low or too high, the outcome may shift from beneficial to toxic, contingent on the specific conditions. These results align with the significance of miRNAs in fine-tuning gene networks. Consequently, we propose that miRNAs play a pivotal role in adjusting the levels of components of the PDI pathway, thereby controlling redox balance and oxidative stress response, while also potentially affecting ER stress.

Figure R9. *hsp-4* promoter activity upon genetic (*pdi-2* RNAi) or chemical (15 μ M bepristat) inhibition of *C. elegans* PDI.

Figure R10. Survival on paraquat of wild type control (WT - ctrl) or *alg-1* overexpressor (ALG-1/OE) worms exposed to 15 μ M bepristat or vehicle. Representative results of 3 replicates.

Figure R11. Lifespan of wild type control (WT - ctrl) or *alg-1* overexpressor (ALG-1/OE) worms exposed to 15 μ M bepristat or vehicle. Representative images of 3 replicates.

Figure R12. Survival on sodium arsenite of wild type control (WT - ctrl) worms exposed to 15 μ M bepristat or vehicle. Representative images of 2 replicates.

Figure R13. Survival on paraquat, sodium arsenite and longevity of wild type (WT) worms exposed to *pdi-2* or control (ctrl) RNAi.

(xiv) Fig. 6: Since the PDI pathway is closely linked to ER function and proteostasis, the above-mentioned reference (PubMedID 34723149, 2021) is particularly relevant to mention here. The paper anticipates the current findings to a certain while limited extent.

R: In alignment with the Reviewer's recommendation, we have incorporated the reference and relevant discussion into the updated manuscript. We sincerely apologize for the oversight in our initial submission, as previously clarified.

(xv) Fig. 7: Does *alg-1* overexpression compensate for the reduced longevity of the *skn-1* zu135 strain?

R: We thank the Reviewer for providing the opportunity to delve more deeply into the associations between ALG-1 overexpression and various stress resistance pathways. In response to the Reviewer's suggestion, we silenced *skn-1*, and as recommended by Reviewer #2, we also investigated the impact of *daf-16* knockdown on paraquat resistance induced by ALG-1 overexpression. Given the essential roles of both DAF-16 and SKN-1 in stress resistance and longevity resulting from germline deficiency⁴⁻⁸, and the possibility that they might function downstream of ALG-1, we explored their involvement in the context of ALG-1 overexpression. Additionally, to answer the Reviewer, we examined whether ALG-1 overexpression could

compensate for the reduced longevity caused by *skn-1* loss-of-function. We did not use the *skn-1 zu135* strain because we did not have it available in the lab at the moment. However, *skn-1* RNAi also leads to shorter lifespan (refer to the data below, only to Reviewers).

In the paraquat stress assay, we observed that knockdown of *daf-16* or *skn-1* diminished worm survival, while ALG-1 overexpression increased survival irrespective of *daf-16* and *skn-1* silencing (refer to the data below and new Supplementary Fig. 5c-d of the revised manuscript). This suggests that ALG-1 induction does not rely on these transcription factors to enhance oxidative stress resistance. Consequently, ALG-1 overexpression could partially rescue compromised stress resistance under *skn-1* or *daf-16* RNAi. Contrarily, in the lifespan assay, ALG-1 overexpression does not promote longevity and does not counteract the shorter lifespan caused by *skn-1* RNAi, in agreement with our consistent finding that ALG-1 overexpression does not extend lifespan under unstressed conditions (refer to the data below, only to Reviewers).

Considering our primary focus on stress resistance and the inability to confirm SKN-1 as a negative regulator of *alg-1*, as indicated in the initial submission (see response to Reviewer #2), we have reevaluated our model. We conclude that SKN-1 does not directly interact with ALG-1 to control oxidative stress resistance. However, this does not rule out the possibility that SKN-1 indirectly controls ALG-1, possibly via miR-71 (as demonstrated by ⁹), and reciprocally, ALG-1 may indirectly influence SKN-1, potentially through redox signaling. We have made revisions to the manuscript to align with these conclusions and edited the model schematic (Fig. 6) accordingly.

Figure R14. Survival on paraquat of wild type control (WT - ctrl) or *alg-1* overexpressor (ALG-1/OE) worms upon *skn-1* or *daf-16* RNAi.

Figure R15. Lifespan of wild type control (WT - ctrl) or *alg-1* overexpressor (ALG-1/OE) worms upon *skn-1* RNAi.

Reviewer #2:

In this manuscript Vergani et al. dissect the role of ALG-1, the major microRNA Argonaute in *C. elegans*, in stress resistance and lifespan. Prior work established that *alg-1* promotes long lifespan upstream of the *C. elegans* FOXO protein DAF-16. Similar to *daf-16* mutants, the authors found that *alg-1* mutants were sensitive to oxidative stress and expressed markers associated with oxidative stress. Next, the authors used published mRNA-seq data as well as a translational reporter to establish that *alg-1* is overexpressed in four different long-lived mutant backgrounds, most prominently in the germline-less *glp-1* mutant. Starting with modENCODE ChIP-seq data, the authors identified five transcription factors that bind the *alg-1* promoter and negatively regulate the expression of a GFP::*alg-1* transgene. One of these transcription factors, R02D3.7, is the best candidate of a direct regulator of *alg-1* expression because it is downregulated in all of the lifespan mutants where *alg-1* is upregulated. They provide evidence that a second transcription factor, GEI-1, regulates *alg-1* expression indirectly via *mir-71*. To complement the loss-of-function experiments, the authors overexpress *alg-1* and show increased resistance to oxidative stress and upregulation of one oxidative stress marker. The use of two different methods to overexpress ALG-1 greatly strengthens their conclusions, and the transgene overexpression includes a very nice control of a similar transgene that does not express *alg-1*. Consistent with R02D3.7 as a negative regulator of *alg-1*, RNAi of R02D3.7 increased resistance to oxidative stress in an *alg-1*-dependent manner. The authors characterized changes in miRNA levels in *glp-1* mutants and in *alg-1* overexpressing strains, thereby identifying three microRNA mutants that are sensitive to oxidative stress: *mir-87*, *mir-230*, and *mir-235*. Finally, they identified a predicted miR-230 target, *pdi-2*, as a negative regulator of stress resistance.

This paper is interesting, well-written, and thoughtful. The role of *alg-1* and microRNAs in stress resistance has not previously been explored and this paper provides new insight from the regulation of *alg-1* all the way to a particular microRNA target. The data are clear and convincing. The comments below are intended to further strengthen this work. All of the changes I suggest could be addressed in the writing. However, if the authors wish to provide more evidence for their claims, I also mention some examples of possible experiments that could be carried out.

R: We are grateful for the kind comments and positive feedback from the Reviewer, and we are pleased to learn that the Reviewer considers our paper of interest, well-written, and found it to have clear and convincing data. While the Reviewer left room for addressing suggestions in the writing, we also conducted additional experiments to address specific points raised, deeming them pertinent for reaching conclusions that could enhance the robustness of our paper. Below, we offer detailed responses to each of these suggestions.

Major comments:

1. The details of transgenes and CRISPR mutants are not clear to me from the methods, Supplemental Table, or text of the paper. It is difficult for me to interpret findings from these tools without an understanding of what specific sequences are present in these strains. If the transgenes and CRISPR mutants are derived from published work, please cite the relevant papers. If any of them were constructed for this paper, please provide the details in the Methods section.

R: We apologize if the information was not conveyed clearly in the manuscript. All strains utilized in our study were generated by other researchers, and we have cited the original studies where

they were first described. To enhance clarity, we have modified the text to better explain how the strains were generated and included a schematic in Supplementary Fig. 3 illustrating the various strains employed for ALG-1 overexpression. Additionally, we have revised the language to explicitly state that these strains were generated by others and have been previously characterized.

2. The experiment in Fig 6B showed that miR-230 can target the *pdi-2* 3' UTR in mammalian cell culture, but it does not show that *pdi-2* is a true target of miR-230 in *C. elegans*. The authors can either adjust their language accordingly or perform the experiment in *C. elegans*, for example by tagging the endogenous *pdi-2* locus and examining expression in a miR-230 mutant vs. wild-type. If desired, the argument could be further strengthened by mutating the miR-230 sites in the *pdi-2* 3' UTR.

R: While we acknowledge the Reviewer's concern, it is important to highlight that 3'-UTR reporter assays using mammalian cells are widely regarded as the gold-standard method in the literature for demonstrating direct targeting of mRNAs by miRNAs^{10,11}. This system offers the advantage of being reductionist, minimizing variables that could indirectly affect the readout and potentially lead to ambiguous results. In this specific context, mammalian cells do not naturally express *cel-miR-230-3p*, ensuring that any alterations in luciferase activity result exclusively from the exogenous miRNA introduced into the cell targeting the *C. elegans pdi-2* 3'-UTR. The incorporation of mutant 3'-UTR in our assays provides additional evidence, reinforcing the specificity of the observed targeting. On the other hand, conducting the suggested experiment in *C. elegans* may introduce additional complexities, as other factors could potentially interfere with *pdi-2* levels indirectly. That considered, we acknowledge that proof of direct targeting in a mammalian system does not automatically translate to *in vivo* targeting, given the dependence on factors such as co-expression of the miRNA and the target, among others. Considering that and the Reviewer's concern, we revised the language of the manuscript to tone down the statements, where we now propose that *pdi-2* can be directly targeted by miR-230.

3. *daf-16* is well characterized to promote resistance to oxidative stress and the reason the authors provide for investigating the role of *alg-1* in oxidative stress is the known regulation of *daf-16* by *alg-1*. However, the interplay of *daf-16* and the new factors characterized here is not considered. The authors provide arguments in the discussion (lines 369-387) that *alg-1* can regulate stress resistance and longevity independently of *daf-16* and this argument seems reasonable as a general statement, but no data are provided to specifically address this point. The authors could either present possibilities in the discussion or perform experiments, for example testing the ability of reducing *pdi-2* function to suppress oxidative stress sensitivity of *daf-16* mutants or examining the expression of miR-230 and *pdi-1* in *alg-1* mutants vs. *daf-16* mutants.

R: We appreciate the opportunity provided by the Reviewer to delve deeper into the interactions between ALG-1 and DAF-16. Furthermore, in response to the comments from Reviewer #1, we explored the involvement of SKN-1, another stress response transcription factor, in the mechanisms by which ALG-1 overexpression protects from oxidative stress. To directly address the Reviewers' points, we conducted knockdown experiments for both *daf-16* and *skn-1*, evaluating whether ALG-1 relies on these transcription factors to enhance oxidative stress

resistance. The results of these experiments, assessed in the paraquat stress assay, revealed that knockdown of *daf-16* or *skn-1* reduced worm survival, while ALG-1 overexpression increased survival regardless of *daf-16* or *skn-1* silencing (refer to the data below and new Supplementary Fig. 5c-d of the revised manuscript). This indicates that ALG-1 induction does not depend on these transcription factors to enhance oxidative stress resistance. Consequently, ALG-1 overexpression could partially rescue compromised stress resistance under *skn-1* or *daf-16* RNAi. These findings provide additional experimental evidence supporting the notion that *alg-1* overexpression can promote oxidative stress resistance independently of *daf-16*, and show that these effects also occur independently of *skn-1*. While it is conceivable that *alg-1* loss-of-function may compromise the activity of DAF-16 (as demonstrated by others¹) and/or SKN-1, it is important to note that even if ALG-1 influences these transcription factors, their individual roles alone are insufficient to fully explain why ALG-1 overexpression provides protection from oxidative stress, suggesting additional mechanisms are in place. In the manuscript, we propose these mechanisms involve miRNAs that can target and fine-tune the PDI pathway. We have incorporated the new results into the manuscript and discussed them accordingly.

Figure R14. Survival on paraquat of wild type control (WT - ctrl) or *alg-1* overexpressor (ALG-1/OE) worms upon *skn-1* or *daf-16* RNAi.

4. From what I could see in Supplemental Table 6, it appears there are two different GFP::*alg-1* reporters: One I assume from the “Is” name to be an exogenous multicopy transgene with the *alg-1* promoter, GFP fused to the N-terminus of ALG-1, and the *alg-1* 3’UTR. The other reporter is the endogenous *alg-1* allele tagged with GFP, also at the N-terminus. Why was the screen for transcription factors that regulate *alg-1* expression was done with the exogenous transgene? The findings would appear to be more meaningful in the endogenous context. If the authors want to strengthen the arguments in Fig 2 and Fig S1, key experiments could be repeated in the endogenous context.

R: The Reviewer correctly points out the distinction in genotypes between the two ALG-1 reporter strains used in the study and their respective methods of generation. While we initially utilized the transgenic line for the RNAi screen due to its availability in the laboratory at the time we initiated the experiment, we agree with the Reviewer regarding the advantages of using the CRISPR-engineered strain. Hence, in response to this suggestion, we employed the CRISPR-engineered strain to validate the positive hits identified in the initial screen performed with the transgenic strain (refer to the data below and new Fig. 1f of the revised manuscript). Our findings confirmed that GEI-2/MEP-1, NHR-77, and R02D3.7 indeed negatively regulate *alg-1* expression in both reporter strains used to assess ALG-1 levels. However, it is noteworthy that while *nhp-28* or *skn-1* RNAi led to the derepression of *alg-1* in the strain generated by transgene insertion, it did not impact ALG-1 levels in the strain generated by CRISPR engineering. This discrepancy suggests that RNAi targeting *nhp-28* or *skn-1* may be influencing ALG-1 levels in the transgenic

strain through unspecific mechanisms, such as the inhibition of transgene silencing, for instance. We express our gratitude to the Reviewer for proposing this experiment, which has significantly contributed to the robustness of our manuscript.

Figure R16. Relative GFP fluorescence after RNAi exposure using the CRISPR-engineered ALG-1 reporter strain where GFP was fused to the N-terminus of endogenous ALG-1. Combined data of three independent experiments.

Minor points:

1. It would be helpful to specify that the “ctrl” RNAi is empty vector in the figure legends, so the reader does not need to search the methods and know what L4440 is to understand the experiment.

R: We altered the figure legends in accordance with the Reviewer’s suggestion.

2. Please explain the term “OP50-1”, as opposed to the more commonly used “OP50” bacteria.

R: OP50-1 is the streptomycin resistant strain of OP50. We explained it in the Methods section.

3. It is interesting that both *alg-1* RNAi and overexpression of *alg-1* cause increased expression of *sod-3p::GFP* (Fig. 1D & Fig 3E). Consider including a brief comment explaining why this may be the case.

R: Considering that Reviewer #1 also raised this point, we decided to perform an experiment to investigate it with more details. To explain this, we hypothesized that the increase in *sod-3* promoter activity could reflect a persistent state of oxidative stress induced in worms subjected to *alg-1* RNAi. Supporting this hypothesis, our data show that both paraquat exposure and *alg-1* RNAi increase *sod-3* promoter activity to a comparable extent, with no additive effects

observed (see Supplementary Fig. 1c). Nonetheless, to directly test this hypothesis, we conducted a new experiment employing the antioxidant N-acetylcysteine (NAC). NAC effectively suppressed the elevated *sod-3* promoter activity in worms exposed to *alg-1* RNAi, but not in worms overexpressing *alg-1* (refer to new Fig. 2i and below). Moreover, in worms overexpressing *alg-1*, paraquat further increased *sod-3* promoter activity levels (see Fig. 2g). These findings support the notion that the activation of the *sod-3* promoter in *alg-1* deficient worms is a consequence of endogenously produced pro-oxidants, indicative of an inherent state of oxidative stress. Conversely, in worms overexpressing *alg-1*, the upregulation of the *sod-3* promoter occurs independently of the redox state, or at least remains insensitive to pro-oxidants mitigated by NAC. This suggests a direct regulatory mechanism that protects worms from oxidative stress when ALG-1 is overexpressed. The manuscript has been updated to incorporate these supporting new data, accompanied by discussion of the results.

Figure R2. *sod-3* promoter activity upon *alg-1* RNAi or overexpression in worms treated with N-acetylcysteine (NAC) or vehicle.

References

1. Aalto, A. P. *et al.* Opposing roles of microRNA Argonautes during *Caenorhabditis elegans* aging. *PLoS Genet* **14**, e1007379 (2018).
2. Finger, F., Ottens, F. & Hoppe, T. The Argonaute Proteins ALG-1 and ALG-2 Are Linked to Stress Resistance and Proteostasis. *MicroPubl Biol* **2021**, (2021).
3. Bekendam, R. H. *et al.* A substrate-driven allosteric switch that enhances PDI catalytic activity. *Nat Commun* **7**, 12579 (2016).
4. An, J. H. & Blackwell, T. K. SKN-1 links *C. elegans* mesendodermal specification to a conserved oxidative stress response. *Genes Dev* **17**, 1882–1893 (2003).
5. Steinbaugh, M. J. *et al.* Lipid-mediated regulation of SKN-1/Nrf in response to germ cell absence. *Elife* **4**, (2015).

6. Lin, K., Hsin, H., Libina, N. & Kenyon, C. Regulation of the *Caenorhabditis elegans* longevity protein DAF-16 by insulin/IGF-1 and germline signaling. *Nat Genet* **28**, 139–145 (2001).
7. Amrit, F. R. G. *et al.* DAF-16 and TCER-1 Facilitate Adaptation to Germline Loss by Restoring Lipid Homeostasis and Repressing Reproductive Physiology in *C. elegans*. *PLoS Genet* **12**, e1005788 (2016).
8. Berman, J. R. & Kenyon, C. Germ-Cell Loss Extends *C. elegans* Life Span through Regulation of DAF-16 by *kri-1* and Lipophilic-Hormone Signaling. *Cell* **124**, 1055–1068 (2006).
9. Smith-Vikos, T. *et al.* MicroRNAs Mediate Dietary-Restriction-Induced Longevity through PHA-4/FOXA and SKN-1/Nrf Transcription Factors. *Current Biology* **24**, 2238–2246 (2014).
10. Koscianska, E. *et al.* Prediction and preliminary validation of oncogene regulation by miRNAs. *BMC Mol Biol* **8**, 79 (2007).
11. Martin, H. C. *et al.* Imperfect centered miRNA binding sites are common and can mediate repression of target mRNAs. *Genome Biol* **15**, R51 (2014).

Reviewer #1 (Remarks to the Author):

The authors have performed all additional experiments requested by the reviewer. In most cases they have now appropriately addressed the previously remaining questions.

Reviewer #2 (Remarks to the Author):

In this revised manuscript, all of my concerns have been thoroughly addressed. I appreciate the detailed, thoughtful response provided by the authors.